   

# Client recruitment mechanism of the cytosolic Fe-S cluster assembly targeting complex

Wenjie Ren [ID][1,2], Yuxin Huang [ID][1,2], Min Hu[1], Yanyang Yang [ID][1], Wen Yang [ID][1✉] & Hui Wang [ID][1✉]

## Abstract

**Most cytosolic and nuclear eukaryotic Fe-S proteins acquire their critical Fe-S cofactor by interacting with the cytosolic Fe-S cluster assembly targeting complex (CTC). Despite the critical roles these Fe-S proteins play in fundamental biology, how they are specifically recognized by the CTC remains largely understudied. Here we identified a hidden consensus pentapeptide motif as a sequence signature dictating cluster acquisition in a majority of known human Fe-S proteins, particularly DNA/RNA processing enzymes for genome maintenance. The presence of this motif drives CTC-client engagement, while its defect impairs CTC recognition, iron incorporation, and enzymatic activities of these clients, ultimately compromising their cellular functions, such as in DNA repair. Furthermore, our studies revealed a conserved surface pocket of CTC dedicated to client recruitment in general. This single pocket recognizes two distinct sequence signatures in clients including the Pentapeptide motif and a previously reported C-tail motif. Subsequent structure-guided affinity-purification mass spectrometry (AP-MS) enabled us to investigate the pocket-dependent human CTC interactome, potentially unveiling unrecognized Fe-S proteins. Overall, our findings decipher the sequence signature-directed mechanism underlying CTC client recruitment and open an avenue for expanding the repertoire of Fe-S proteins.**

**Keywords** Cluster Acquisition; CTC; DNA Repair; Fe-S Proteins; Pentapeptide Motif
**Subject Categories** Post-translational Modifications & Proteolysis; Structural Biology; Translation & Protein Quality

## Introduction

Fe-S clusters are ancient, inorganic prosthetic group essential for virtually all organisms from bacteria to humans, mediating reactions in life-sustaining processes such as respiration, photosynthesis, and nitrogen fixation (Lill, 2009; White and Dillingham, 2012). As modular cofactors with diverse chemical properties, Fe-S clusters readily accept and donate electrons. They are considered as optimal redox mediators and are crucial in the aforementioned

activities (Barton et al, 2019; Beinert et al, 1997; Johnson et al, 2005) by sensing and responding to redox disturbances. Studies of Fe-S proteins date back to the 1960s (Beinert et al, 1997). Approximately 35 years ago, Fe-S clusters were discovered in an enzyme involved in DNA repair (Cunningham et al, 1989). Since then, more and more of these cofactors have been identified in proteins responsible for DNA and RNA metabolism. These include, but are not limited to, DNA helicases XPD, FANCJ, and RTEL1, DNA helicase-nuclease DNA2, DNA primase, DNA glycosylase MUTYH, DNA polymerases α, δ and ε, tRNA methylthiotransferase CDKAL1, and tRNA 4-demethylwyosine synthase TYW1, underscoring their significance in maintaining genome stability (Arragain et al, 2010; Fuss et al, 2015; Klinge et al, 2007; Netz et al, 2011; Rudolf et al, 2006). In fact, genetic alterations that impair the biogenesis and activity of Fe-S proteins have been linked to a growing number of distinct human diseases, such as Friedreich's ataxia (FRDA), xeroderma pigmentosum, Fanconi anemia, and cancers (Maio and Rouault, 2022; Rouault and Tong, 2008).

Despite the well-established roles of Fe-S proteins in human health and disease, the known inventory of Fe-S proteins remains surprisingly small, largely due to the unique nature of Fe-S clusters. As ancient cofactors essential for sustaining life processes, a good portion of Fe-S clusters are firmly integrated into the holoproteins and well-protected. However, in the modern oxygen-rich environment, certain Fe-S clusters, particularly those in regulatory proteins, are highly prone to oxidative destruction, leading to issues during the production and purification of recombinant Fe-S proteins (Imlay, 2006; Vallieres et al, 2024). As a result, Fe-S proteins are challenging to isolate in their intact form. Moreover, the binding ligands of Fe-S clusters do not follow a well-defined pattern and are often unpredictable (Pritts and Michel, 2022; Vallieres et al, 2024), further making Fe-S proteins elusive for identification and characterization using conventional methods. The lack of detection curbs exploring the pool of Fe-S proteins that may be crucial to genomic integrity and human health. This highlights the compelling need for alternative approaches to identify previously unrecognized and hidden Fe-S proteins, presumably by decoding common features of these proteins to effectively extract them from the proteome for further validation.

In search of such common features, we focus on the process by which Fe-S clusters are delivered to their apo-proteins. In human cells, Fe-S clusters are biosynthesized in mitochondria before being delivered to target cytoplasmic and nuclear apo-proteins for incorporation (Dancis et al, 2024; Lill and Freibert, 2020). Despite

[1]Greater Bay Biomedical InnoCenter, Shenzhen Bay Laboratory, Shenzhen 518132, China. [2]These authors contributed equally: Wenjie Ren, Yuxin Huang.
✉E-mail: yangwen@szbl.ac.cn; hwang@szbl.ac.cn

the longstanding controversy over the parallel Fe-S cluster biogenesis in subcellular compartments other than mitochondria (Maio and Rouault, 2020), this process is understood to occur through a series of highly regulated and coordinated steps, facilitated by specialized mitochondrial and cytosolic machineries (Lill et al, 2015; Netz et al, 2014; Rouault, 2015). Biogenesis is initiated when the cysteine desulfurase NFS1 abstracts sulfur from cysteine and transfers it onto a scaffold protein, where an Fe-S cluster is built de novo by incorporating concurrently received iron ion and electrons. The nascent Fe-S clusters are subsequently handled by collaboratively working carrier/chaperone protein complexes for safe escort and presentation to the apo-proteins. Among these protein complexes, a late-acting targeting complex known as cytoplasmic Fe-S assembly targeting complex (CTC), which contains CIAO1, CIAO2B, and MMS19, is required for the insertion of Fe-S clusters into the majority of apo-clients, as the last step of cluster delivery (Gari et al, 2012; Stehling et al, 2012). Functional and systematic proteomic studies have shown that, during Fe-S cluster delivery, the CTC is able to engage with, although in slightly different binding specificity and complex composition, approximately 30 cytosolic and nuclear client proteins. These clients, once cluster-incorporated, play roles in processes ranging from DNA replication and repair, telomere maintenance, translation, and tRNA modification to amino acid and nucleotide metabolism (Netz et al, 2014; Stehling et al, 2013). Despite the growing understanding of the key players and clients in Fe-S cluster acquisition over the past decade, it remains unclear how ONE CTC recognizes all the client proteins with specificity, regardless of their functional and structural diversity. The possibility that the mechanism behind this recognition involves common features or sequence signatures in clients driving their interaction with the CTC hasn't been fully explored.

An earlier study proposed that a lysine-tyrosine-arginine (LYR) tripeptide motif in a subset of Fe-S proteins aids in cluster delivery through an early-acting cochaperone protein, HSC20 (Maio et al, 2014). This suggests that a sequence feature in clients could also dictate cluster acquisition from the late-functioning CTC. Besides, it has been recognized in the field that a tryptophan or phenylalanine (W/F) containing C-terminus of CTC clients is critical for their cluster acquisition (Marquez et al, 2023; Paul et al, 2015; Upadhyay et al, 2014). However, only one-third of known human CTC clients bear such C-termini, leaving the recognition mechanism for most of CTC clients unsolved. We herein delved into the recognition between CTC and its known clients in humans, by means of integrated biochemical, structural, cellular, and proteomics analyses, to uncover a consensus pentapeptide motif in a majority of known Fe-S proteins engaged in genome maintenance, as a sequence signature that determines the CTC-client recognition. This motif is essential for the cluster acquisition and functionality of these clients, particularly in DNA repair, within cells. It leads to the revelation of a pocket in the CTC designated for sequence signature-directed client recruitment and sheds lights on the discovery of hidden Fe-S proteins.

## Results

### Fragments in the N-terminal region of CDKAL1 drive CTC recognition

To interrogate the recognition between human CTC and its clients, we set out to reconstitute their interactions in recombinant purified

system. Despite the notoriously finicky purification of known Fe-S proteins, we successfully isolated a few of them, conducted a brief screening, and chose to focus on CDKAL1, an enzyme involved in tRNA modification (Landgraf et al, 2016) with relatively good behavior and clear domain annotation, for mapping the CTC recognition in depth.

As a client, CDKAL1 displays decent and comparable binding affinities with both the complete CTC and its subcomplex containing CIAO1 and CIAO2B, as demonstrated in pull-down and quantitative binding assays. By contrast, neither MMS19 nor CIAO1 alone can recognize CDKAL1 (Figs. 1A,B and EV1B,D). We hence utilized the CIAO1-CIAO2B subcomplex as the core CTC (Fig. EV1A) to pinpoint the CTC-interacting region in CDKAL1 hereafter. According to the domain description, we made constructs of CDKAL1 covering different regions and analyzed their binding with the CTC. To our surprise, besides the full-length CDKAL1, only CDKAL1ΔC and CDKAL1-N can be pulled down (Figs. 1C and EV1C), indicating that the N-terminal flexible region of CDKAL1, not the central catalytic region loaded with Fe-S clusters, is recognized by the CTC. We further conjugated the sequence of CDKAL1-N to a CTC-irrelevant protein, ubiquitin, and successfully transformed it into a CTC interactor (Figs. 1C and EV1C), thereby reinforcing the role of CDKAL1-N in CTC interaction.

To nail down and identify the CTC recognition site, we performed the alanine scanning mutagenesis on the basis of CDKAL1-N. Residues 1–55 in CDKAL1 were sequentially substituted in blocks of five by alanine to generate a series of CDKAL1-N variants, namely A1 to A11 (Fig. 1D). All the variants were overexpressed and tested for their binding activities against the CTC. As shown in Fig. 1D, when compared with the WT and other variants, the variants A3, A4, A8, and A9, which carry alanine substitution of fragment 3 (F3) at residue 11–15, fragment 4 (F4) at residue 16–20, fragment 8 (F8) at residue 36–40, and fragment 9 (F9) at residue 41–45, respectively, lost the binding activities substantially, demonstrating the importance of fragments covering residues 11–20 (F3–F4) and residues 36–45 (F8–F9) in CTC physical engagement. To confirm our observations, we did quantitative CTC binding analyses using proteins of CDKAL1-N, and fragments containing residues 1–45 (F1–F9), residues 1–35 (F1–F7), and residues 21–45 (F5–F9) (Figs. 1E and EV1E). Interestingly, the absence of F8–F9 in the fragment containing residues 1–35 reduced the binding affinity by nearly 15 folds, while the fragment comprising residues 1–45 retained full competence for CTC binding, exhibiting an affinity similar to that of CDKAL1-N at ~200 nM. However, without F3 and F4, the fragment containing residues 21–45 (F5–F9) completely lost the CTC engagement. We thus biochemically mapped the CDKAL1-CTC recognition, which is primarily driven by a region spanning residues 11–20 (F3–F4) of CDKAL1, with additional support from another region covering residues 36–45 (F8–F9).

### CDKAL1-CTC interface is structurally defined by a conserved pentapeptide in CDKAL1

We next aimed to understand the molecular basis of CDKAL1-CTC recognition. To this end, we conducted a structural study of the core CTC, consisting of the heterodimeric CIAO1-CIAO2B, in complex with CDKAL1-N or its fragments containing the key

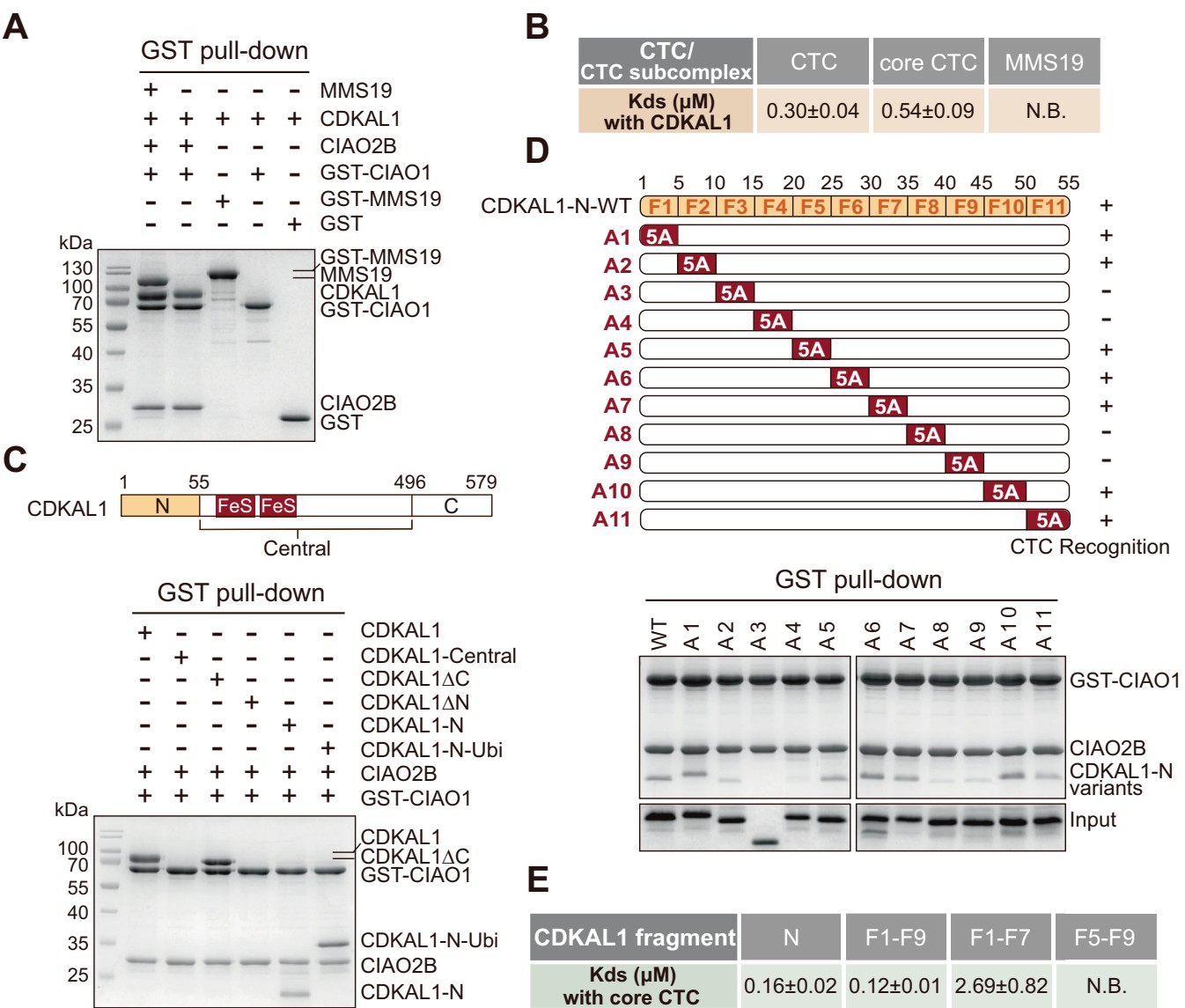

**Figure 1. Biochemically mapping the CTC–CDKAL1 recognition.**

(A) GST pull-down assay using GST-tagged CTC, its subcomplex and components and purified CDKAL1 with GST as a negative control. See also Fig. EV1B for proteins as input. (B) The binding affinities of CDKAL1 to the CTC and its subcomplex determined by ITC quantitatively. N.B.: No detectable binding activity. See also Fig. EV1D for ITC binding curves. (C) Domain organization of CDKAL1 and GST pull-down assay using GST-tagged core CTC and purified constructs of CDKAL1 covering different regions. See also Fig. EV1C for proteins as input. (D) Alanine scanning mutagenesis testing the binding activities of CDKAL1-N variants against the core CTC by GST pull-down assay. The design of alanine substitution variants on the basis of CDKAL1-N was shown in the upper panel. (E) The binding affinities of fragments in CDKAL1-N to the core CTC determined by ITC quantitatively. N.B.: No detectable binding activity. See also Fig. EV1E for ITC binding curves. Source data are available online for this figure.

determinant regions. After rounds of unsuccessful crystallization trials with various complex combinations, we compromised to determine, at a resolution of 2.5 Å, the structure of a pentapeptide within human CDKAL1-N F3–F4 (residues 11–20) in complex with *Drosophila* CIAO1-CIAO2B (PDB: 9UKY, Figs. 2A and EV2B, Table EV1). The *Drosophila* version shares over 70% sequence identity with the human counterpart and binds human CDKAL1-N F3–F4 with a similar affinity (Fig. EV2A, in structural analysis, the amino acid numbering of CIAO1-CIAO2B is based on the *Drosophila* version while that of CDKAL1 is in human version).

The overall architecture of *dm*CIAO1-CIAO2B presents as previously reported (Kassube and Thoma, 2020), with CIAO1 adopting a typical WD40-repeat β-propeller fold. It interacts with the small globular CIAO2B, mainly its helices α3 and α4, through the tips of β-propeller blades 2, 3 and 4 on the narrow top side (Fig. EV2B). CDKAL1 pentapeptide is found sitting at the cleft forged from the CIAO1-CIAO2B interface, extensively sandwiched by the CIAO1 β-propeller tips and the CIAO2B helical folds. Comprising residues D12 to I16, CDKAL1 pentapeptide (hereafter referred to as pentapeptide) is intimately involved in core CTC

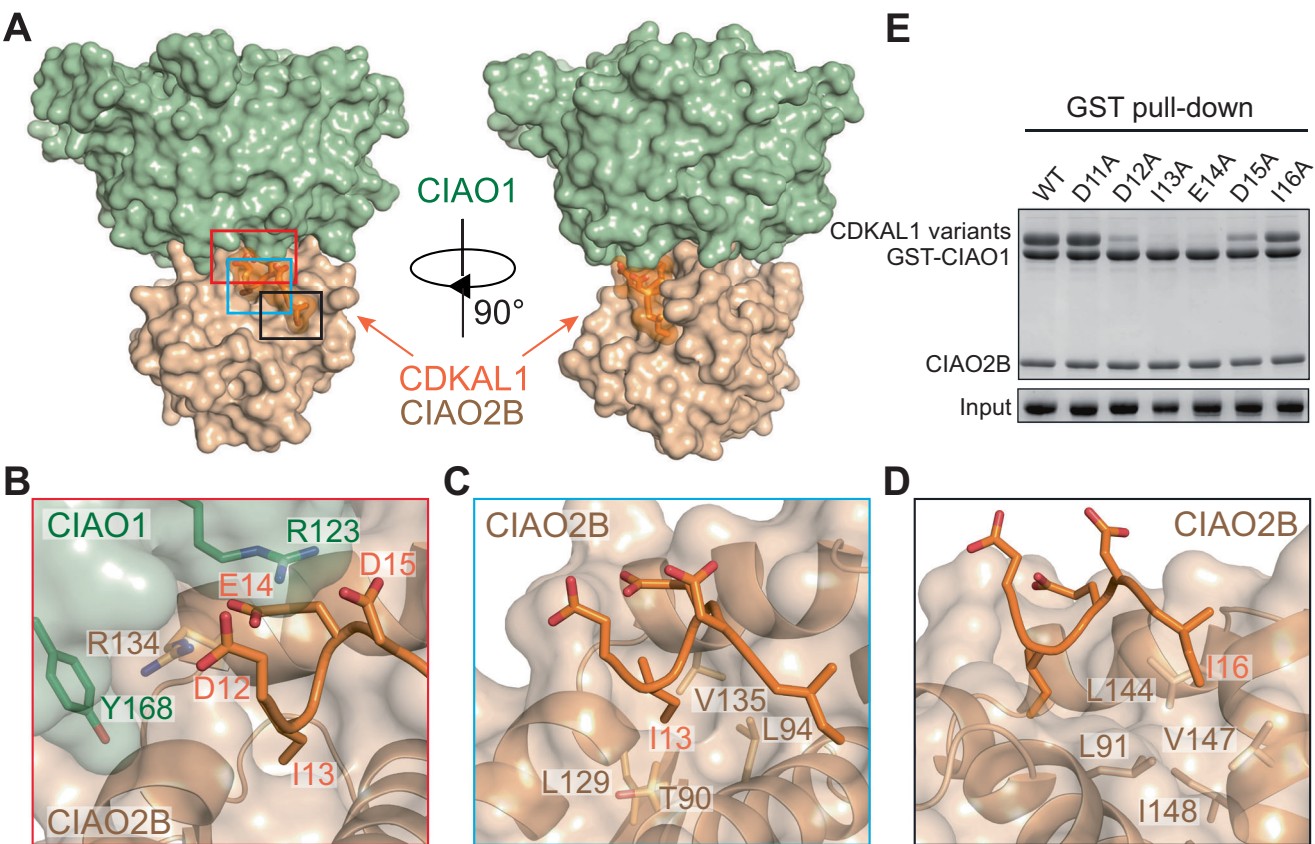

**Figure 2. Structural revelation of the CDKAL1-CTC interface.**

(**A**) Two orthogonal views of the complex comprising *dm*CIAO1-CIAO2B and *h*CDKAL1 pentapeptide shown in surface diagram with CIAO1 in sage green, CIAO2B in wheat, and CDKAL1 pentapeptide in orange, respectively. The pentapeptide is also shown in ribbon with side chains in sticks inside the transparent surface diagram. (**B–D**) Close-up views of the interface between CDKAL1 pentapeptide and *dm*CIAO1-CIAO2B complex featuring (**B**) charged interactions and (**C, D**) two hydrophobic patches. CIAO1 and CIAO2B are shown in surface and ribbon with the same color scheme as shown in (**A**), and CDKAL1 pentapeptide is shown in orange ribbon. Select interface residues are shown in sticks. (**E**) GST pull-down assay assessing the binding abilities of CDKAL1 WT and variants regarding the pentapeptide and its flanking region toward human CIAO1-CIAO2B complex. Source data are available online for this figure.

recognition and directly participates in the physical engagement with the CIAO1-CIAO2B complex (Fig. 2A). We superposed the pentapeptide-bound CIAO1-CIAO2B structure with the solved structure of CTC containing CIAO1-CIAO2B and MMS19 (Kassube and Thoma, 2020) (PDB: 6TC0). As one would reasonably conceive, MMS19 associates with the core CTC at a site opposite to where the pentapeptide binds, which explains the dispensable role of MMS19 in the CDKAL1-CTC recognition, as disclosed by our biochemical assays (Fig. EV2C).

A close examination of the pentapeptide binding area reveals three pivotal spots that propel the recognition predominantly. First, three acidic residues in pentapeptide, D12, E14, and D15, are riveted onto the cleft between CIAO1 and CIAO2B by interacting with positively charged arginine residues from CIAO1 blade 3, R123, and from CIAO2B α4 helix, R134. In particular, the pentapeptide E14 side chain is firmly positioned due to the bipartite charge-charge interactions from both arginine residues. A cation-π interaction between CIAO2B R134 and CIAO1 Y168, which apparently becomes a part of the CIAO1-CIAO2B interface, further stabilizes R134 and the above charged network (Fig. 2B). Besides acidic residues, two hydrophobic residues in pentapeptide,

I13 and I16, are captured and secured, respectively, by two hydrophobic cavities formed exclusively by the residues from CIAO2B helical regions. I13 is well accommodated in a deep and major cavity made of residues T90, L94, L129, and V135, while I16 resides in a shallow and minor hollow area provided by residues L91, L144, V147, and I148 (Fig. 2C,D). Hydrophobic interactions regarding both I13 and I16 reinforce the pentapeptide-CTC recognition. In addition, a body of contacts targeting the backbone and side chain of the pentapeptide, which includes CIAO2B Q128, N141, and CIAO1 K125, help to maintain a peripheral environment favorable for recognition (Fig. EV2D).

To probe the importance of pentapeptide in upholding human CTC engagement, according to our structural analyses, we tested a set of CDKAL1 variants with alanine substitution of individual amino acid in pentapeptide and its flanking region (as a control) for CTC binding (Fig. 2E). As to the residues involved in the charged network, replacement of D12 and D15 considerably affected the binding, suggesting a necessary role of these residues at the interface. By contrast, CTC interaction was completely abolished by the substitution of E14, manifesting its essentiality in the CTC recognition. This finding aligns with the high stringency of this

residue observed in our structure. For hydrophobic residues in pentapeptide, alteration of I13 abolished the binding utterly, bolstering its essentiality as equal to that of E14. Meanwhile, the effect of I16 mutation is marginal with slightly compromised binding activity. As expected, the sequence of this pentapeptide is conserved across CDKAL1 orthologs from humans to placozoa and archaeon (Fig. EV2E), further fortifying its significance in potentially mediating a critical bioprocess, such as CTC recognition. On the other hand, when we aligned the fragment containing CDKAL1-N F8–F9 across the same species, we observed relatively low conservation, which supports the supplementary role of this fragment (Fig. EV2F). Interestingly, both fragments are missing in bacterial CDKAL1 orthologs (Arragain et al, 2010), consistent with the absence of the CTC complex and the reliance on alternative mechanisms for Fe-S cluster biogenesis and insertion in these species (Braymer et al, 2021; Frazzon and Dean, 2003). Taken together, we structurally revealed a CDKAL1-CTC interface defined by a conserved pentapeptide in CDKAL1.

## The pentapeptide represents a consensus motif for CTC recognition in Fe-S proteins

The presence of a conserved pentapeptide in CDKAL1 dictating CTC recognition, along with the observed stringency and tolerance of this pentapeptide at the interface interacting with the CTC, suggests that this pentapeptide may serve as a consensus binding motif for CTC recognition in Fe-S proteins. To thoroughly analyze the pentapeptide as a binding motif and precisely determine the consensus signature of each amino acid within this peptide, we employed isothermal titration calorimetry (ITC) in combination with mutagenesis of amino acid at every single position to measure the binding affinities of these pentapeptide variants, in the context of the CDKAL1-N fragment, toward the core CTC (Fig. 3A; Appendix Fig. S1). At position 1, substitutions of D to its acidic analog E, neutral analog N, and hydrophobic residue A all decreased the affinities by 10 folds or more, demonstrating D as the favorite residue. In the following position which features hydrophobic association with the CTC, mutations to either the small residue A or the bulky residue F destroyed the recognition, while hydrophobic residues with medium-sized side chain are able to maintain the binding and tolerated at this position. The residue E at position 3 is the most stringent within the pentapeptide and cannot be replaced by any other residues, including its acidic analog D. This emphasizes the unique geometry of its side chain for CTC recognition. The third acidic residue involved in the charged network at position 4 is more permissive than the first two. Both acidic analog E and neutral analog N of the parental D were shown to be the competent substitutes, however, hydrophobic residues and polar residues with unfavorable geometry were poorly tolerated. In the final position, which also features hydrophobic interactions, residue I can be altered to any amino acids except charged and large polar residues like E, R, and Q, without losing too much binding activity. This observation goes along with the modest impact of I to A mutation on disrupting recognition revealed by the interface analysis. Collectively, the pentapeptide identified in CDKAL1, if taken as a putative generic binding motif for the CTC, possesses a consensus sequence of $D_1$-$[I/V/L/M]_2$-$E_3$-$[D/E/N]_4$-{charged and large polar residues}$_5$, which has been validated both structurally and biochemically (Fig. 3B).

Next, we searched the pool of known human nuclear and cytosolic Fe-S proteins that are CTC clients for this consensus motif. To our excitement, we found that 10 of these proteins, primarily key DNA/RNA processing enzymes for genome maintenance, harbor the motif without ambiguity. Particularly, FBXL5, an Fe-S protein previously identified and characterized (Wang et al, 2020), contains this motif in a region that is reportedly involved in the physical interaction between FBXL5 and the CTC (Mayank et al, 2019). An alignment of sequences carrying this consensus pentapeptide, along with the corresponding sequence logo, confirms the presence of the motif and illustrates its conservation within a collection of Fe-S proteins (Fig. 3C), backing its significance to these proteins. To further validate the role of this consensus pentapeptide motif in mediating cellular CTC-client recognition, we substituted the first four amino acids of the pentapeptide with alanine, creating motif-deficient mutants of these Fe-S proteins. We then assessed their direct interactions with the CTC by co-IPs in HEK293T cells. As shown in Fig. 3D, compared to the WT proteins, all the mutants lost their binding activities toward the CTC dramatically, if not in entirety, highlighting the intimate involvement of the pentapeptide with consensus sequence in CTC-client recognition in known Fe-S proteins. In light of our findings, we name this consensus motif in Fe-S proteins, as a primary CTC recognition site, the Pentapeptide.

## Pentapeptide defect impairs iron incorporation and enzymatic functions of Fe-S proteins

To assess the role of the Pentapeptide in determining the cluster acquisition and functionality of Fe-S proteins containing it, we isolated, to the best of our ability, some of the overexpressed Fe-S proteins from HEK293F cells in both WT and pentapeptide-deficient forms (as described above). We then tested their iron incorporation and corresponding DNA processing activities in vitro.

Cluster acquisition of an Fe-S protein is commonly monitored by its iron incorporation. Anaerobically prepared proteins of WT and variants of FANCJ, DNA2, MUTYH, TYW1, and CDKAL1 were analyzed using a colorimetric assay to determine their iron content for comparison, as previously documented (Maio et al, 2021). We found that, compared with the control protein containing a fully loaded [2Fe-2S] cluster (2 iron atoms per protein), all tested WT Fe-S proteins expressed in mammalian cells bound iron in the range of 1.3–2.1 moles/mole of purified protein, lower than the expected ratio. By contrast, variants lacking the proper pentapeptide to interact with the CTC bound significantly less iron (Fig. 4A). Meanwhile, complementary measurement of acid-labile sulfide via classic methylene blue assay yielded values of $1.634 \pm 0.072$ and $0.425 \pm 0.037$ moles of sulfide per mole of protein for CDKAL1 WT and its mutant, respectively, which are comparable to their respective iron-loading data. This insufficiency of Fe-S cluster incorporation may arise from the limited cluster loading capacity of the mammalian cells utilized in our experiments, as well as partial oxidative damage during isolation. Consistently, the UV/Vis spectra of WT proteins displayed a bump of characteristic absorbance at 420 nm, whereas the pentapeptide-deficient variants showed featureless spectra (Fig. EV3A). Together, these results revealed that pentapeptide deficiency impairs iron incorporation and, consequently, disrupts cluster acquisition.

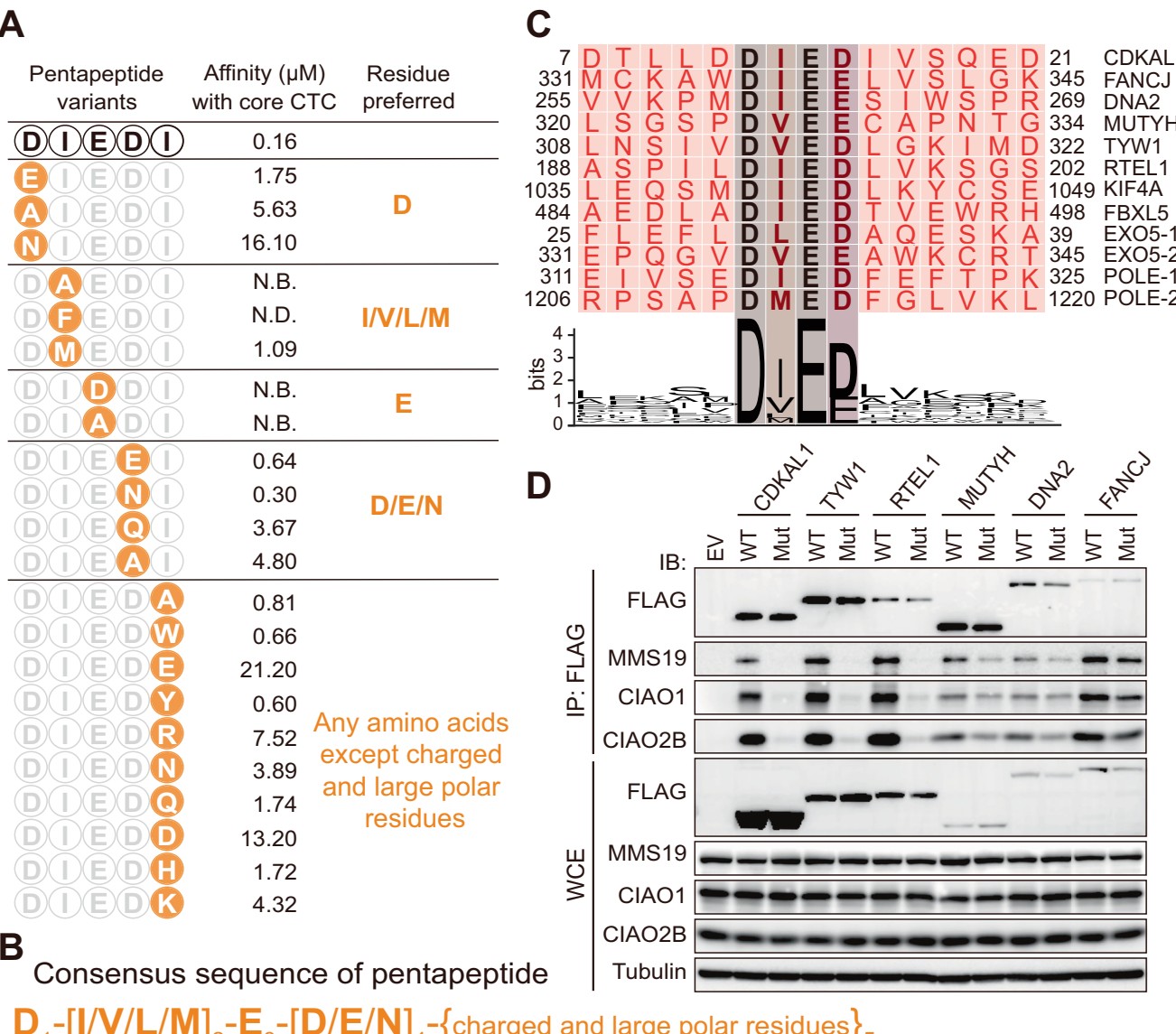

**B** Consensus sequence of pentapeptide

$$D_1\text{-}[I/V/L/M]_2\text{-}E_3\text{-}[D/E/N]_4\text{-}\{\text{charged and large polar residues}\}_5$$

**Figure 3. The pentapeptide as a consensus motif for CTC-binding in Fe-S proteins.**

(**A**) The binding affinities of pentapeptide variants to the core CTC determined by ITC quantitatively. Residue substitutions in pentapeptide were indicated by orange circle and the preferred residues for each position were highlighted in orange. N.B.: No detectable binding activity. N.D.: Not determined due to the weak binding. See also Appendix Fig. S1 for ITC binding curves. (**B**) A consensus sequence of pentapeptide summarized from (**A**) is shown in orange. Indicated position of amino acid in pentapeptide is shown as subscript. A paired square bracket '[]' represents a position where any residues in the bracket is accepted, while a paired curly bracket '{}' represents a position of any amino acids except for those denoted inside. (**C**) Sequence alignment of the consensus pentapeptide motif found in known human CTC client Fe-S proteins. For comparison, the flanking sequences are also shown at both sides. Two consensus pentapeptide motifs in both EXO5 and POLE1 are identified and labeled accordingly. Residues are colored from red to black according to their conservations from low to high. A corresponding sequence logo is generated by WebLogo (Crooks et al, 2004) and presented beneath the alignment with all residues colored in black. The overall height of a stack indicates the sequence conservation at that position, while the height of symbols within the stack indicates the relative frequency of each amino acid at that position. The most conserved four residues in the pentapeptide motif are highlighted. (**D**) HEK293T cells were transiently transfected with FLAG-tagged indicated Fe-S proteins in forms of either WT or consensus motif-deficient mutant. Whole cell extracts (WCE) were subjected to immunoprecipitations with anti-FLAG beads and immunoblotting against the antibodies of FLAG, MMS19, CIAO1 and CIAO2B. The transfected RTEL1 was RTEL1[1-767aa] in mouse version, and the FANCJ was FANCJ[1-1100aa]. EV: empty vector. Source data are available online for this figure.

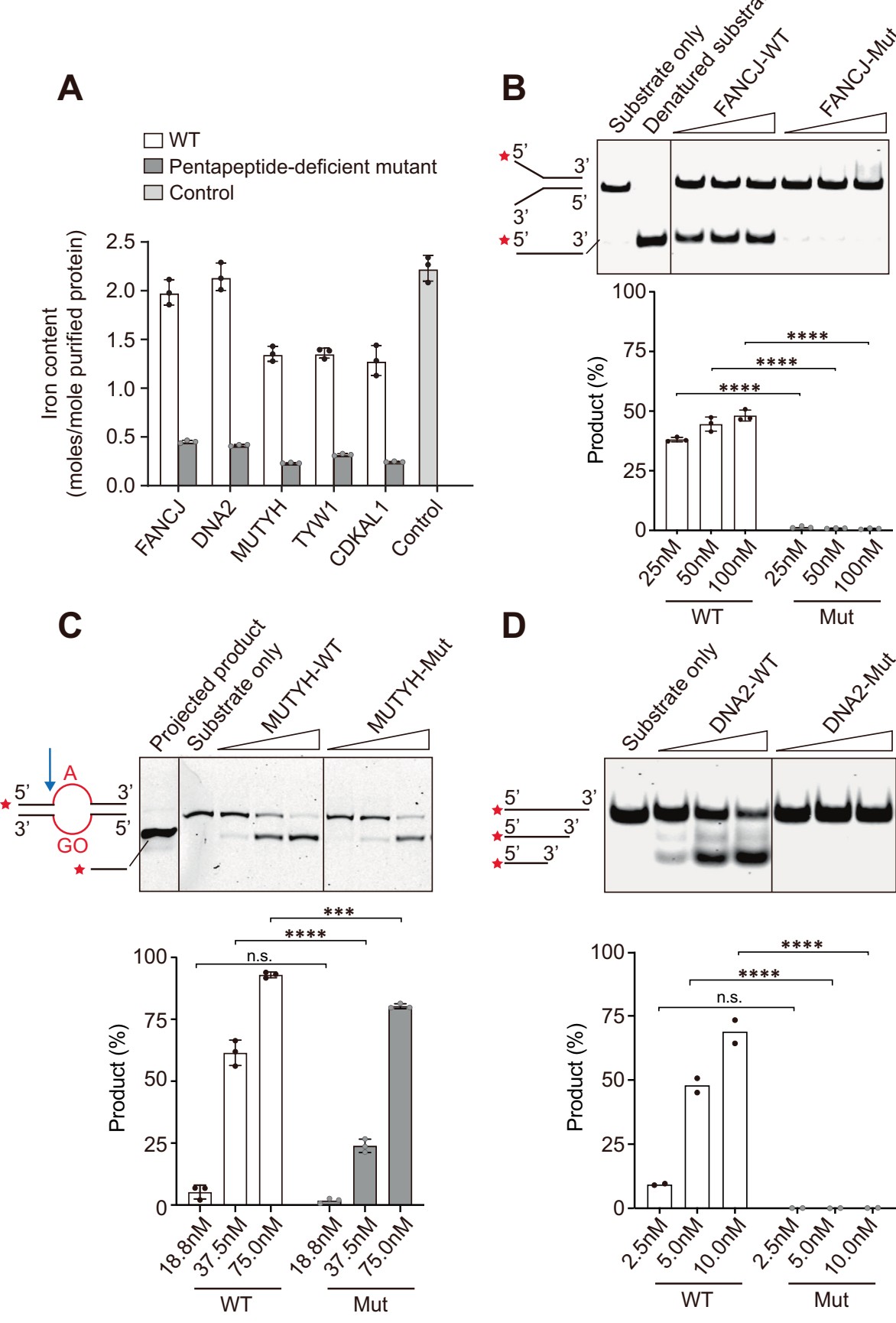

◄  **Figure 4.   Pentapeptide affects cluster acquisition and functionality of Fe-S proteins in vitro.**

(A) Colorimetric assay determining the iron content of indicated FLAG-tagged Fe-S proteins WT and their pentapeptide-deficient mutants. Levels are expressed as moles/ mole purified protein. Error bars depict standard deviation of the mean from three technical replicates. The purified FBXL5-SKP1-IRP2 complex is used as a control. (B) DNA helicase assay with $^{cy5}$Y-substrate comparing 25 nM, 50 nM, 100 nM FLAG-FANCJ$^{1-1100aa}$ WT and 25 nM, 50 nM, 100 nM FLAG-FANCJ$^{1-1100aa}$ mutant. Substrate only without proteins and heat-denatured substrate indicating the unwound product are shown as controls. Quantification (Mean ± SD) of FANCJ's DNA helicase activity showing the fraction of DNA product calculated from three technical replicates. ****$p < 0.0001$ ($p = 7.3e-12$ for 25 nM, $p = 9.45e-13$ for 50 nM and $p = 3.52e-13$ for 100 nM) (two-way ANOVA). (C) Adenine DNA glycosylase assay with $^{cy5}$dsDNA$_{39mer}$ containing A:GO mismatch (GO represents 8-oxo-G), comparing 18.8 nM, 37.5 nM, 75 nM FLAG-MUTYH WT and 18.8 nM, 37.5 nM, 75 nM FLAG-MUTYH mutant. Synthesized projected product and substrate only without proteins are shown as controls. Quantification (Mean ± SD) of MUTYH's adenine DNA glycosylase activity showing the fraction of DNA product calculated from three technical replicates. ****$p < 0.0001$ ($p = 3.06e-09$), ***$p < 0.001$ ($p = 0.0003$), n.s.: not significant ($p = 0.3760$) (two-way ANOVA). (D) DNA nuclease assay with $^{cy5}$ssDNA$_{42nt}$ comparing 2.5 nM, 5 nM, 10 nM FLAG-DNA2 WT and 2.5 nM, 5 nM, 10 nM FLAG-DNA2 mutant. Substrate only without proteins is shown as control. Quantification (mean with individual data points) of DNA2's nuclease activity showing the fraction of DNA product calculated from two technical replicates. ****$p < 0.0001$ ($p = 1.36e-05$ for 5 nM and $p = 1.61e-06$ for 10 nM), n.s.: not significant ($p = 0.0716$) (two-way ANOVA). Source data are available online for this figure.

Next, the corresponding DNA processing activities of FANCJ, MUTYH and DNA2 were tested. As to the DNA helicase FANCJ, the DNA unwinding assays were performed in the presence of ATP and oligo-based Y-structure DNA as a substrate (Gupta et al, 2005) (Figs. 4B and EV3B). Correlating with its reduced ability to acquire iron, the pentapeptide-deficient FANCJ variant exhibited defective unwinding of the substrate at increasing concentrations (Fig. 4B). The adenine DNA glycosylase activity of MUTYH was analyzed on a 39 bp A:GO mismatch-containing DNA duplex (Bai et al, 2005) (Figs. 4C and EV3C). As opposed to the WT protein, the pentapeptide-deficient MUTYH variant displayed decreased adenine excision activity against the designated substrate (Fig. 4C). In assays evaluating the dual helicase-nuclease activities of DNA2 (Fig. EV3D,E), when the Y-structure DNA substrate was employed, we observed unwound and chopped products from the WT protein. In contrast, the pentapeptide-deficient mutant only produced unwound products (Fig. EV3E), suggesting that, due to impaired Fe-S cluster delivery, the mutant likely lost its nuclease activity while retaining helicase activity to some extent. Further assay with single-strand DNA substrate confirmed an utter loss of nuclease activity (Fig. 4D). Notably, upon checking the solved structure of DNA2 (Zhou et al, 2015) (PDB: 5EAX), we discovered that the Fe-S cluster is bound exclusively to the nuclease domain, leaving the helicase domain cluster-free (Fig. EV3F). This finding is perfectly in agreement with our observations in the assays. Therefore, in the purified system, we demonstrated that pentapeptide failure in Fe-S proteins not only diminishes their iron incorporation but also compromises their activities as DNA processing enzymes.

## Pentapeptide alteration in FANCJ compromises its role in genome maintenance

We next sought to probe the significance of the Pentapeptide in Fe-S proteins when these proteins are functioning against DNA damage in cells to maintain genome stability. The cell-based assessment of the Pentapeptide focused on the multifunctional DNA helicase FANCJ, also known as BRCA1-interacting protein 1 (BRIP1) or BRCA1-associated C-terminal helicase 1 (BACH1). This helicase plays well-established roles in the Fanconi anemia (FA) pathway for DNA inter-strand crosslink (ICL) repair and in DNA double-strand break (DSB) repair via homologous recombination (HR) (Ceccaldi et al, 2016; Litman et al, 2005). We decided to implement a deletion/complementation system to study the ability of pentapeptide-related variants to functionally complement

a FANCJ-deficient cell line. To achieve this, we first generated a HeLa cell line where FANCJ was knocked out using CRISPR/Cas9 technique (FANCJ KO, Fig. EV4A). We then stably complemented the FANCJ KO HeLa cell line with ectopic FANCJ WT and different FANCJ variants, giving rise to a series of cell lines for testing. It is of note that, to obtain a more stable and long-lasting gene product for the pentapeptide-deficient FANCJ variant, we used two milder FANCJ mutants with altered pentapeptide instead of four alanine substitutions. These mutants, E338D (where residue E at position 3 of the pentapeptide is replaced with D, hereafter referred to as M1) and I337A/E338D (where residue I at position 2 and residue E at position 3 are replaced with A and D, respectively, hereafter referred to as M2), have both been proved to be CTC-binding deficient by our biochemical analyses. Besides, the levels of exogenously expressed FANCJ constructs, although comparable to each other, were lower than that of the endogenous FANCJ (Fig. 5A).

Clonogenic survivals were investigated to oversee the cell sensitivity to the ICL-inducing agent mitomycin C (MMC). As previously reported (Boavida et al, 2024; Litman et al, 2005), FANCJ KO cells displayed high sensitivity to MMC, showing pronounced growth restriction (Figs. 5B and EV4B). Complementation with WT FANCJ significantly restored resistance to MMC, though not to the level of the parental HeLa cell line, largely due to the low exogenous expression of WT FANCJ. On the contrary, cells expressing either FANCJ M1 or M2, mutants with revised pentapeptide, could not regain MMC resistance, still holding high sensitivity similar to that of FANCJ KO cells (Figs. 5B and EV4B). These observations denoted that pentapeptide deficiency in FANCJ substantially attenuated its ability to repair DNA ICLs and sensitized the cells to DNA-damaging agents.

In addition to the ICL repair, FANCJ has been reported to participate in DNA DSB repair via HR in association with BRCA1 (Cantor et al, 2001). To test the influence of the Pentapeptide on the HR pathway, we did the canonical HR assay using the DR-GFP reporter (Pierce et al, 1999) (Fig. EV4C). During the assay, the I-SceI endonuclease was transfected to generate DSBs. An internal GFP (iGFP) could repair this break only through effective HR to form functional GFP. Successfully repaired event will give rise to GFP signals detected by flow cytometry. In HeLa-derived cell lines, we discerned that depletion of FANCJ considerably reduced HR rates. Complementation with the WT protein fully restored the deficient HR, whereas the mutants with pentapeptide alterations, FANCJ M1 and M2, failed to do so (Figs. 5C and EV4D).

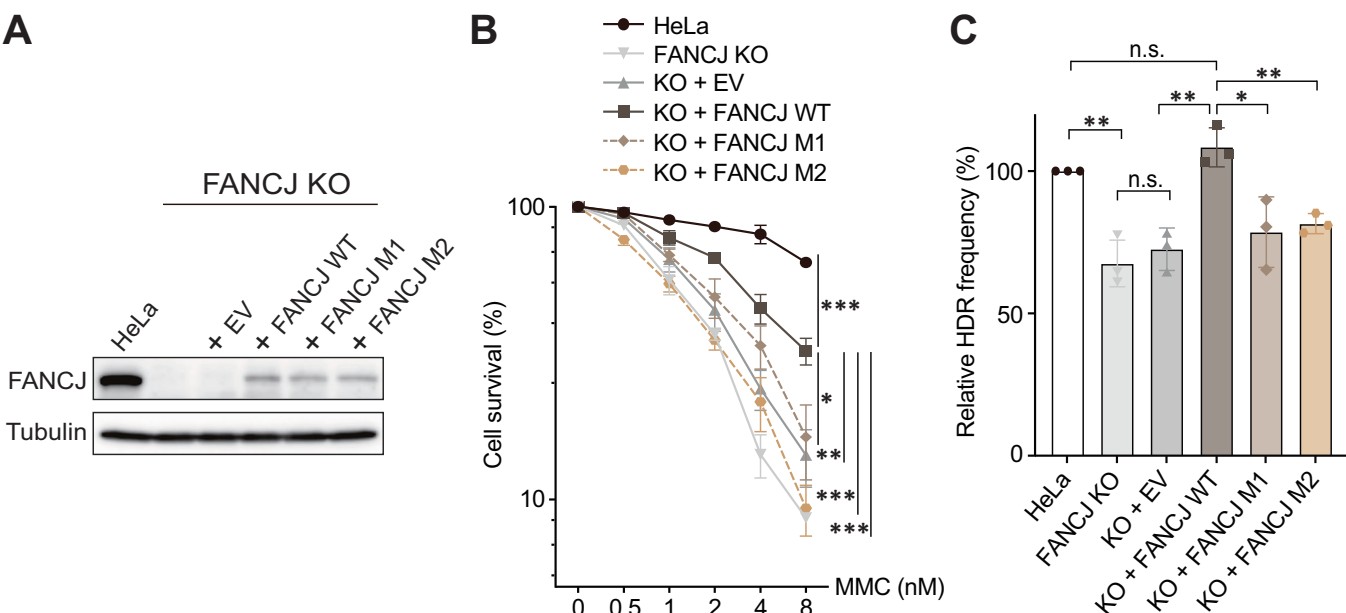

**Figure 5. Pentapeptide deficiency influencing FANCJ functions in cells.**

(A) Immunoblot analysis of HeLa and HeLa-derived cell lines in FANCJ deletion/complementation system via α-FANCJ antibody. EV: empty vector. (B) Survival curves of HeLa and HeLa-derived cell lines in FANCJ deletion/complementation system treated with indicated concentrations of MMC. The mean values (± SEM) from three biological replicates were plotted. ***$p < 0.001$, **$p < 0.01$, *$p < 0.05$ (two-way ANOVA). Exact P values: HeLa vs FANCJ$^{KO}$ + FANCJ$^{WT}$, $p = 0.0009$; FANCJ$^{KO}$ + FANCJ$^{WT}$ vs FANCJ$^{KO}$ + FANCJ$^{M1}$, $p = 0.0287$; FANCJ$^{KO}$ + FANCJ$^{WT}$ vs FANCJ$^{KO}$ + EV, $p = 0.0081$; FANCJ$^{KO}$ + FANCJ$^{WT}$ vs FANCJ$^{KO}$ + FANCJ$^{M2}$, $p = 0.0003$; FANCJ$^{KO}$ + FANCJ$^{WT}$ vs FANCJ$^{KO}$, $p = 0.0002$. EV: empty vector. See also Fig. EV4B for representative images. (C) Quantification of results from HR assays in HeLa and HeLa-derived cell lines in FANCJ deletion/complementation system. The GFP-positive cells indicate the fraction of successfully completed HR events. The mean values (± SD) from three biological replicates were plotted. **$p < 0.01$, *$p < 0.05$, n.s.: not significant (t-test). Exact P values: HeLa vs FANCJ$^{KO}$, $p = 0.0024$; FANCJ$^{KO}$ vs FANCJ$^{KO}$ + EV, $p = 0.4755$; FANCJ$^{KO}$ + EV vs FANCJ$^{KO}$ + FANCJ$^{WT}$, $p = 0.0036$; FANCJ$^{KO}$ + FANCJ$^{WT}$ vs FANCJ$^{KO}$ + FANCJ$^{M1}$, $p = 0.0216$; FANCJ$^{KO}$ + FANCJ$^{WT}$ vs FANCJ$^{KO}$ + FANCJ$^{M2}$, $p = 0.0038$; HeLa vs FANCJ$^{KO}$ + FANCJ$^{WT}$, $p = 0.0998$. EV: empty vector. See also Fig. EV4D for representative FACS plots. Source data are available online for this figure.

In conformance to the results from the clonogenic assays, the lack of a proper pentapeptide compromised FANCJ's role in the HR pathway for fixing DNA DSBs. Altogether, as exemplified in the case of FANCJ, we established an intimate role of the Pentapeptide in Fe-S proteins in controlling DNA repair within cells, potentially influencing the genome stability.

## A revealed CTC client recruitment pocket potentiates expanding the pool of Fe-S proteins

Considering our discovery of the Pentapeptide, a consensus motif in a subset of Fe-S proteins that defines CTC-client recognition to mediate the functionalities of these proteins in DNA/RNA processing, we wondered the correlations between our findings and the previously reported C-terminal W/F involving recognition mechanism (Marquez et al, 2023), in terms of the physical sites of client recruitment in the CTC. To address this question, we first validated the C-terminal W/F mechanism in humans by co-IPs (Fig. EV5A) and then generated a binding model of this recognition by AlphaFold 3 (Abramson et al, 2024) using RSAD2 as a client in alignment with our structure (Fig. 6A). Remarkably, both the Pentapeptide and C-tail W/F motifs interact with the CTC through a shared pocket, mimicking each other's polar and hydrophobic network (Figs. 6A and EV5B). This pocket is nestled in a concentrated region between CIAO1 and CIAO2B. We individually mutated three conserved key residues, R125 from human CIAO1,

and Q135 and R141 from CIAO2B, within the pocket and observed a loss of or dramatically reduced binding toward both subsets of clients (Fig. 6B). It is of note that, in support of our findings, the yeast counterpart of human CIAO1 R125 has been reported to be important for CIAO1's function regarding the cytosolic Fe-S protein assembly (Srinivasan et al, 2007). Interestingly, consistent with the comparable binding affinities of the pentapeptide-dependent client CDKAL1 and the C-tail-dependent client RSAD2 to the core CTC complex quantified by ITC (Figs. EV1D and EV5C), a bead-based biochemical competition assay showed that, even with marginal passive dissociation of the client from the pre-assembled CTC-client complex during extensive washing, CDKAL1 and RSAD2 were able to compete with each other reciprocally for CTC binding in an active manner, whereas pentapeptide-deficient CDKAL1 completely lost its competitive capacity (Figs. 6C and EV5D,E). These observations reinforce the shared pocket as the designated site for client recruitment in the CTC.

Inspired by the revelation of the CTC client recruitment pocket, we hypothesized that a defective pocket would, in principle, eliminate the interactions between the CTC and its cellular Fe-S protein clients. To test this idea, we produced two versions of pocket-deficient CTC, enclosing CIAO1-R125A and CIAO2B-R141A, respectively, as baits and conducted affinity pull-down to examine their interacting proteome alongside the CTC-WT. As shown in Fig. 6D, the two pocket-deficient CTC baits unveiled a surprisingly similar pattern of the pulled-down products on the gel

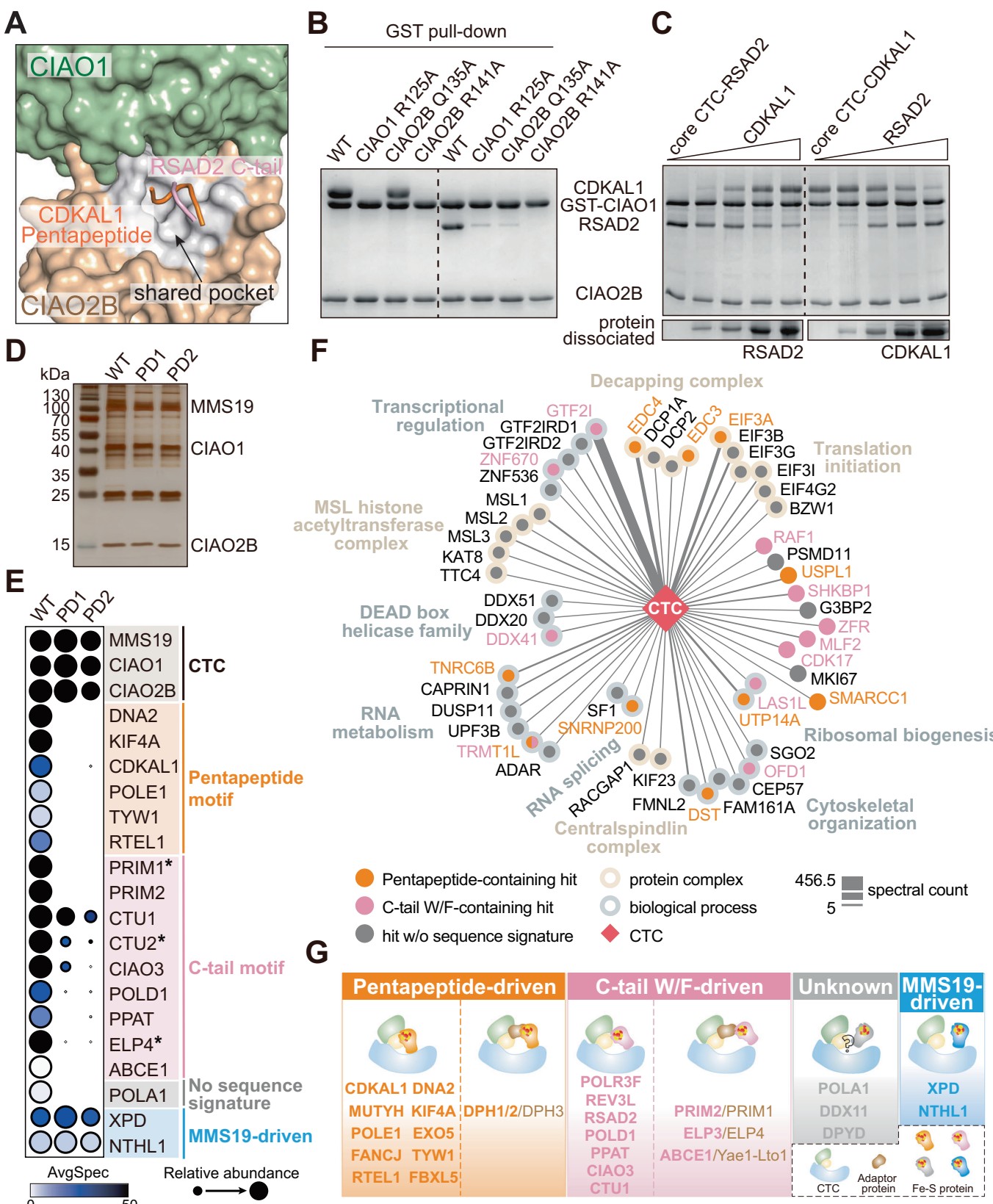

◀

**Figure 6.  A CTC client recruitment pocket for exploring hidden Fe-S proteins.**

(A) Close-up view of a structural model showing a shared pocket in CTC for client recruitment generated by superposition of the structure of *dm*CIAO1-CIAO2B-*h*CDKAL1 pentapeptide and the binding model of CTC-C-tail motif recognition. The binding model is created by AlphaFold 3 using RSAD2 as a client. For clarity, only the last three residues at the C-terminus of RSAD2 are shown. CIAO1 and CIAO2B are shown in surface with CIAO1 in sage green and CIAO2B in wheat. CDKAL1 pentapeptide and RSAD2 C-tail are shown in ribbon with CDKAL1 pentapeptide in orange and RSAD2 C-tail in pink. The area of shared pocket is highlighted in light gray. See also Fig. EV5B for interaction details. (B) GST pull-down assay assessing the binding abilities of the core CTC in forms of WT and pocket-deficient variants toward the clients of CDKAL1 and RSAD2. (C) In vitro competition assays demonstrating the abilities of both CDKAL1 and RSAD2 to compete with each other for the CTC binding reciprocally. Dissociation of the pre-bound clients was monitored in the flowthrough. (D) Representative SDS-PAGE gel of affinity pull-down using the CTC in forms of WT and pocket-deficient variants (PD1: Pocket-deficient 1, containing CIAO1-R125A, PD2: Pocket-deficient 2, containing CIAO2B-R141A) as baits. The pulled down proteins against HEK293T cell lysate were detected by silver staining. (E) Dot plot generated in ProHits-viz program visualizing the average spectral counts of known human Fe-S proteins captured in the AP-MS by using the CTC WT and pocket-deficient variants PD1 and PD2 as baits. Fill color and size of dots represent AvgSpec and Relative abundance, respectively. The fill color represents the average spectral count of each protein, with darker shading indicating higher counts (maximum capped at 50; values above 50 are shown in black). The dot size reflects the relative abundance of each protein across the three datasets. CTC components, Pentapeptide-containing Fe-S proteins, C-tail-containing Fe-S proteins, Fe-S proteins with no sequence signature but pocket dependent, and pocket independent MMS19-driven Fe-S proteins are grouped, labeled and colored accordingly. Star (*) indicates the adapter or the associated protein of the Fe-S protein. PD1: Pocket-deficient 1, containing CIAO1-R125A, and PD2: Pocket-deficient 2, containing CIAO2B-R141A. (F) Network of PPIs showing the high-confidence CTC-interactors retrieved from Fig. EV5F visualized with Cytoscape. Edge thickness proportional to spectral counts. The CTC is shown as red diamond, and protein hits are shown either individually (circles) or grouped as protein complexes and biological processes (framed circles) according to their reported functions. The Pentapeptide-containing and C-tail-containing proteins are colored in orange and pink, respectively. TRMT1L harbors both motifs and is shown in bicolor. The protein complexes and biological processes are labeled and colored accordingly. (G) A schematic summary of mechanisms underlying the cluster acquisition of Fe-S proteins via the CTC. Pocket-dependent mechanisms comprising Pentapeptide-driven (shown in orange) and C-tail-driven (shown in pink) recognition represent the predominant CTC-client recognition strategies, while a few exceptions utilize either pocket-independent mechanism probably driven by MMS19 (shown in blue) or mechanism still to be investigated (shown in gray). In the schematic boxes illustrating pocket-dependent mechanisms, the left cartoon depicts direct recruitment of a client by the CTC, while the right cartoon represents recognition mediated by an adapter protein. Source data are available online for this figure.

by silver staining. Meanwhile, the CTC-WT captured more distinct protein bands, indicating an apparent loss in the CTC interactome attributed to the pocket deficiency. To unfold this observed difference specifically, we subjected the products of the affinity pull-down to MS analysis. As expected, the majority of known CTC Fe-S protein clients, containing either the Pentapeptide or the C-tail motif, along with their associated proteins, were absent from the pocket-deficient spectra (Fig. 6E; Dataset EV1), reinforcing that the pocket is indeed shared by both motifs and providing proof-of-concept for our proposal. We therefore believe that this differential AP-MS approach will aid to bring to light hidden Fe-S proteins that are pocket-dependent. Proteins or protein complexes exhibiting a spectral counting pattern resembling that of a known CTC client are likely to contain unrecognized Fe-S cluster and may be CTC clients as well. According to this theory, we handpicked the CTC-interacting hits localized to the cytoplasm or nucleus, whose spectral counting was clearly diminished under pocket-deficient conditions compared to the WT. These hits are presented in Fig. EV5F and Dataset EV1. We showcased interactions between the CTC and the interesting hits based on the biological processes they participate in or the functional complexes they belong to (Fig. 6F). Specifically, these included transcriptional regulation, decapping complex, translation initiation, MSL histone acetyltransferase complex, RNA metabolism, ribosomal biogenesis, centralspindlin complex, RNA splicing, DEAD box helicase family, and cytoskeletal organization. The essentiality of these processes and complexes highlighted the potential significance of the Fe-S proteins yet to be disclosed. Moreover, it is noteworthy that, with the exception of the MSL histone acetyltransferase complex and centralspindlin complex, most of the hits, whether individually or in groups, feature the Pentapeptide, the C-tail motif, or both (as in TRMT1L) as CTC recognition site. This further validates the applicability of our findings in broadening the repertoire of Fe-S proteins.

## Discussion

In this study, we identified the Pentapeptide motif in a subset of human Fe-S proteins as a sequence signature to dictate Fe-S cluster acquisition from the cytosolic delivery machinery CTC. Additionally, we uncovered a distinct pocket in the CTC for sequence signature-directed client recruitment. We believe that we've solved the puzzling mechanism underpinning the cluster acquisition of Fe-S proteins via the CTC (Fig. 6G). Among the approximately 30 known CTC client Fe-S proteins, over 80% are pocket-dependent and carrying either the Pentapeptide or C-tail motif as sequence code for their cluster acquisition, rendering the Pentapeptide- and C-tail-dependent mechanisms as the predominant CTC-client recognition strategies. NTHL1 has pocket-independent mechanism possibly driven by MMS19, as previously observed for XPD (Ito et al, 2010; Odermatt and Gari, 2017). Meanwhile, two exceptions, DDX11 and DPYD, obtain Fe-S clusters through unknown mechanism. Although cluster acquisition is primarily triggered by recognition between the sequence signatures in Fe-S proteins and the designated pocket within the CTC, successful delivery often requires extra supports of orienting clients with structural diversity toward an environment favorable for cluster reception (e.g., a distal supplementary fragment of CDKAL1-N for CTC-CDKAL1 recognition, and MMS19 as binding necessity for a subset of Fe-S proteins (Gari et al, 2012; Stehling et al, 2012)), thus providing with adaptable, client-specific accommodations during cluster delivery by the CTC. In line with this notion, structural mapping of the pentapeptide onto Fe-S proteins revealed that, regardless of the common assumption that CTC interacts with Fe-S proteins through their cluster-ligated regions (van Wietmarschen et al, 2012), not all Fe-S cluster binding sites reside in the vicinity of the pentapeptide motifs, some even far off. This suggests that, to adapt to client-specific structural variations and ensure an effective cluster delivery, the CTC may operate through a more spatially dynamic mechanism. However,

the details of how this spatial coordination facilitates cluster transfer remained to be elucidated.

Given the deep-rooted significance of Fe-S proteins in maintaining genome stability (Paul and Lill, 2015; Weon et al, 2018), we utilized the well-studied DNA helicase FANCJ to probe the role of the Pentapeptide in mediating the functionalities of Fe-S proteins. Specifically, we concentrated on testing its interaction with the CTC, iron incorporation, enzymatic activity, and capacity to mediate DNA repair in multiple pathways. Our results clearly demonstrated that, both in vitro and in cells, a defective pentapeptide that depletes CTC recognition led to a flawed Fe-S cluster delivery and thereby compromised FANCJ's helicase activity along with its corresponding DNA repair ability toward ICLs and DSBs via HR to keep genome integrity. It has been documented that genetic alterations affecting Fe-S cluster biogenesis contribute to the loss-of-function of cluster-containing proteins and are causally linked to a growing body of human diseases (Maio et al, 2024; Maio and Rouault, 2022; Rouault and Tong, 2008; van Karnebeek et al, 2024). Our findings give rise to an intriguing possibility that, genetic mutations in CTC clients disrupting Fe-S cluster delivery, like those failing cluster assembly, may also be implicated in human diseases, promising a potential target for therapeutic exploration. Strikingly enough, FANCJ L340F, a mutation found in patients with familial breast cancer and likely to be pathogenic (Kim et al, 2016), occurs precisely at the last position of the pentapeptide in FANCJ. This mutation has been shown to have a partial loss of Fe-S cluster binding and a reduced ability to unwind DNA (Odermatt et al, 2020). In our biochemical mutagenesis binding assays probing the pentapeptide consensus sequence, we observed that substituting the last position of the pentapeptide with a bulky aromatic phenylalanine analog decreased binding affinity toward the CTC by roughly threefold. This result is consistent with the defects previously reported for this mutation (Odermatt et al, 2020) and manifests the causal involvement of Fe-S cluster acquisition deficiency in human disorders, including cancer.

After deciphering the Pentapeptide in Fe-S proteins, we were anxious to look for proteins that possess either this sequence or the previously reported C-tail W/F in human proteome (using ScanProsite (Sigrist et al, 2013) searching against UniProtKB/Swiss-Prot) in hopes of uncovering hidden Fe-S proteins. This search yielded over 2,400 hits, many of which were apparently non-Fe-S proteins, underscoring that sequence signatures alone are insufficient for identifying new Fe-S proteins. To pinpoint the Fe-S protein-to-be more effectively and precisely, we proceeded to reveal a designated pocket in the CTC for general client recruitment and conducted pocket-directed differential AP-MS analysis. To our surprise, this strategy served as a brightly designed experimental filter and effectually narrowed the hits down to no more than 50, creating a compact yet more robust and integrated list of candidates for further validation. Some proteins and protein complexes among them, such as FMNL2, USPL1, MSL complex, and decapping complex, have been shown as CTC interactors in previous proteomics studies (Stehling et al, 2013; Stehling et al, 2012), strongly implying these hits to accommodate Fe-S cluster with high confidence. On top of that, we did sequence analysis regarding the residues that constitute the pocket for client recruitment at the CIAO1-CIAO2B complex interface and found that these residues are conserved across species from humans to yeast, indicating that

the pocket-dependent client recruitment mechanism for Fe-S cluster acquisition is likely conserved throughout evolution. The C-tail W/F sequence signature has been resolved in yeast (Marquez et al, 2023), encouraging us to examine the available orthologs of human Fe-S proteins. Our analysis disclosed that nearly all of the orthologs possess either the Pentapeptide or the C-tail motif, demonstrating that these sequence signatures are conserved across species as well. We therefore postulate that, from humans to yeast, our findings of the sequence code and the CTC pocket recognition mechanism might be exploited as a conserved determinant for Fe-S protein identification. Specifically, our pocket-directed differential AP-MS analysis strategy could allow the discovery of unrecognized Fe-S proteins specific to species beyond humans.

# Methods

### Reagents and tools table

| Reagent/Resource | Reference or Source | Identifier or Catalog Number |
|---|---|---|
| **Experimental models** | | |
| HeLa (*H. sapiens*) | MeilunBio | CL0215 |
| HEK293T (*H. sapiens*) | MeilunBio | CL0169 |
| HEK293FT (*H. sapiens*) | MeilunBio | CL0175 |
| HEK293F (*H. sapiens*) | Gift from Mingjie Zhang lab | N/A |
| HeLa-FANCJ$^{KO}$ (*H. sapiens*) | This study | N/A |
| HeLa-FANCJ$^{KO}$-EV (*H. sapiens*) | This study | N/A |
| HeLa-FANCJ$^{KO}$-FANCJ$^{WT}$ (*H. sapiens*) | This study | N/A |
| HeLa-FANCJ$^{KO}$-FANCJ$^{M1}$ (*H. sapiens*) | This study | N/A |
| HeLa-FANCJ$^{KO}$-FANCJ$^{M2}$ (*H. sapiens*) | This study | N/A |
| **Recombinant DNA** | | |
| pET-28a-CDKAL1 WT, mutants and variants | This study | N/A |
| pET-28a-RSAD2 | This study | N/A |
| pET-GST-CIAO1 human and Drosophila | This study | N/A |
| pET-GST-MMS19 | This study | N/A |
| pAL-CIAO2B human and Drosophila | This study | N/A |
| pET-FBXL5-SKP1 | (Wang et al, 2020) | N/A |
| pFB-GST-IRP2 | (Wang et al, 2020) | N/A |
| pET-StrepII-CIAO1 | This study | N/A |
| pcDNA3.0-FANCJ WT and mutant | This study | N/A |
| pcDNA3.0-DNA2 WT and mutant | This study | N/A |
| pcDNA3.0-MUYTH WT and mutant | This study | N/A |
| pcDNA3.0-CDKAL1 WT and mutant | This study | N/A |

| Reagent/Resource | Reference or Source | Identifier or Catalog Number |
|---|---|---|
| pcDNA3.0-TYW1 WT and mutant | This study | N/A |
| pcDNA3.0-RTEL1 WT and mutant | This study | N/A |
| pLVML-3×FLAG-FANCJ WT and mutants | This study | N/A |
| pCBASceI | MiaoLingPlasmid | P5449 |
| pDRGFP | MiaoLingPlasmid | P5460 |
| psPAX2 | Gift from Zhenghao Li | N/A |
| pMD2.G | Gift from Zhenghao Li | N/A |
| pSpCas9(BB)-2A-GFP-sgRNAs (targeting FANCJ) | This study | N/A |
| **Antibodies** | | |
| anti-CIAO1 | Cell Signaling Technology | 81376 |
| anti-MMS19 | Santa Cruz Biotechnology | sc-390658 |
| anti-CIAO2B | Cell Signaling Technology | 86302 |
| anti-β-Tubulin | Cell Signaling Technology | 15115 |
| anti-DYKDDDDK Tag | Cell Signaling Technology | 14793 |
| anti-FANCJ | Cell Signaling Technology | 4578 |
| anti-rabbit IgG, HRP-linked antibody | Cell Signaling Technology | 7074 |
| anti-mouse IgG, HRP-linked antibody | Cell Signaling Technology | 7076 |
| **Oligonucleotides and other sequence-based reagents** | | |
| 5′-cy5-labeled 42mer oligonucleotide | This study | Methods |
| 5′-cy5-labeled 39mer oligonucleotide | This study | Methods |
| sgRNAs targeting FANCJ | This study | Methods |
| Genotyping PCR primers | This study | Methods |
| **Chemicals, Enzymes and other reagents** | | |
| DMEM | Gibco | 11965092 |
| FBS | Sigma-Aldrich | F0193 |
| Penicillin-Streptomycin | Gibco | 15140122 |
| Trypsin | Gibco | 25200072 |
| Ferric ammonium citrate | Sangon Biotech | A500061-0250 |
| Ferric chloride | Sangon Biotech | A600454-0500 |
| Zinc acetate | Sangon Biotech | A421997-0010 |
| L-cysteine | Sangon Biotech | A600132-0100 |
| N,N-Dimethyl-p-phenylenediamine dihydrochloride | Sigma-Aldrich | D4139 |
| Sodium sulfide | Macklin | S888711 |
| OPM-293 CD05 Medium | OPM | 81075-001 |
| Lipofectamine 3000 | Invitrogen | L3000015 |

| Reagent/Resource | Reference or Source | Identifier or Catalog Number |
|---|---|---|
| Puromycin | Beyotime Biotechnology | ST551-50mg |
| Polyethylenimine | FUSHENBio | FSF0002 |
| Universal Mycoplasma Detection Kit | Yeasen Biotechnology | 40601ES20 |
| Protease Inhibitor Cocktail | Sigma-Aldrich | P8340 |
| Iron Assay Kit | Sigma-Aldrich | MAK025 |
| Pierce Silver Stain Kit | Thermo Scientific | 24612 |
| ATP | MCE | HY-B2176 |
| Proteinase K | Sigma-Aldrich | V900887 |
| Formamide | Sangon Biotech | A501904 |
| Ponceau S | Beyotime Biotechnology | P0022-120ml |
| SuperSignal West Pico PLUS Chemiluminescent Substrate | Thermo Scientific | 34577 |
| Cyanine5.5 NHS ester | Lumiprobe | 27020 |
| Crystal violet | Sangon Biotech | A600331 |
| mitomycin C | Selleck | S8146 |
| DFO | Selleck | S5742 |
| **Software** | | |
| MicroCal PEAQ-ITC Analysis Software v1.40 | https://www.malvernpanalytical.com/en | N/A |
| HKL2000 | https://hkl-xray.com/ | N/A |
| CCP4 suite | https://www.ccp4.ac.uk/ | N/A |
| Coot | https://www2.mrc-lmb.cam.ac.uk/personal/pemsley/coot/ | N/A |
| PHENIX | https://phenix-online.org/ | N/A |
| PyMOL | https://www.pymol.org/ | N/A |
| ImageJ | https://imagej.net/ij/ | N/A |
| GraphPad Prism | https://www.graphpad.com/ | N/A |
| Cytoscape | https://cytoscape.org/ | N/A |
| **Other** | | |
| cOmplete His-Tag Purification Resin | Roche | 5893801001 |
| Glutathione Sepharose 4B | Cytiva | 17075605 |
| Anti-FLAG M2 Affinity Gel | Sigma-Aldrich | A2220 |
| Pierce Anti-DYKDDDDK Magnetic Agarose | Thermo Scientific | A36797 |
| Pierce Streptavidin Magnetic Beads | Thermo Scientific | 88817 |
| HiTrap Q HP | Cytiva | 17115301 |
| Superdex 200 Increase | Cytiva | 28990944 |
| Anaerobic glove box | McCoy | N/A |

## Protein expression and purification

The human Fe-S proteins of CDKAL1 and RSAD2 were expressed as B1 domain of Protein G (GB1) N-terminal fusion proteins and produced in *E. coli* Rosetta (DE3) cells in normal LB media or LB

media supplemented with cysteine and Ferric ammonium citrate (FAC) at the concentrations of 121 mg/L and 25 mg/L, respectively, when needed. The proteins were purified by nickel affinity and subsequent anion exchange (HiTrap Q HP, Cytiva) and size exclusion (Superdex 200 Increase, Cytiva) chromatography after off-column cleavage by tobacco etch virus (TEV) protease. The human heterodimeric CIAO1-CIAO2B complex (core CTC) and the individual components of the CTC, CIAO1, and MMS19, were all produced in *E. coli* BL21 or Rosetta (DE3) cells. CIAO1 and MMS19 were expressed as glutathione S-transferase (GST) N-terminal fusion proteins and isolated by glutathione affinity and subsequent anion exchange (HiTrap Q HP, Cytiva) and size exclusion (Superdex 200 Increase, Cytiva) chromatography after off-column cleavage by TEV protease. The core CTC was obtained by co-expression of GST-tagged CIAO1 and GB1-tagged CIAO2B and purified in the same way as the components. To reconstitute the complete CTC, the isolated proteins of CIAO1-CIAO2B and MMS19 were mixed in stoichiometric amounts and subsequently applied to the size exclusion (Superdex 200 Increase, Cytiva) column for purification. The sample of *Drosophila* CIAO1-CIAO2B for crystallization was prepared in the same way as the human version. All the protein mutants and variants were expressed and isolated in the same way as the WT proteins. The affinity tag may be left on the proteins for the purposes of different assays. Protein of the human FBXL5-SKP1-IRP2 complex was prepared as previously described (Wang et al, 2020).

The human Fe-S proteins utilized in biochemical functional assays in vitro were expressed as FLAG N-terminal fusion proteins and produced in mammalian cells. HEK293F cells were cultured in OPM-293 CD05 Medium at 5% $CO_2$, 37 °C, 100 rpm in the shaker. Plasmid DNA of indicated protein was transfected into the cells using 40 kDa linear polyethylenimines (PEI, FUSHENBio) at 1 µg/ml when the cell density was at $1.0 \times 10^6$/ml. Cells were cultured for another 48 h for protein expression with the supplementation of 40 µM $FeCl_3$ and 300 µM L-cysteine before harvest. Protein isolation was performed in an anaerobic glove box (McCoy). Cells were lysed by mild sonication in lysis buffer (20 mM Tris-HCl pH 7.5, 150 mM NaCl), supplemented with protease inhibitors (Aprotinin, Leupeptin, and Pepstatin A at 1 µg/ml). The insoluble fraction was removed by centrifugation (18,000 rpm) for 90 min at 4 °C. Cleared lysate was incubated with anti-FLAG M2 affinity gel (Sigma-Aldrich) for 2 h and the beads were washed extensively with lysis buffer. FLAG-tagged proteins were eluted by using FLAG peptide at 0.2 mg/ml, concentrated when needed, and analyzed by SDS-PAGE.

### Affinity pull-down assay and competition binding assay

The GST pull-down assay was performed using ~200 µg of purified GST or GST-tagged proteins as the bait and ~500 µg of GB1-tagged iron-sulfur proteins and their mutants and variants. Reaction mixtures were incubated with 100 µl GST beads (Cytiva) at 4 °C for one hour. After extensive wash with binding buffer, the proteins/protein complexes on the beads were eluted by 5 mM glutathione. The eluted samples were resolved by SDS-PAGE and analyzed by Coomassie staining. Inputs represent 3–5% of the total amount of proteins used for each reaction.

In bead-based competition binding assay, purified proteins of GB1-tagged CDKAL1 and RSAD2 were loaded in excess onto the GST beads bound by the core CTC to preassemble the complexes of core CTC-CDKAL1 and core CTC-RSAD2, respectively. After extensive wash with binding buffer, the beads with immobilized complex were aliquoted in a volume of 200 µl (1:1 slurry) with ~200 µg total protein and placed in a panel of 7.5 mm gravity columns. The beads were incubated with 300 µl of purified GB1-tagged RSAD2 or CDKAL1 proteins in a serial dilution (0, 1, 5, 25, and 125 µM) for one hour at 4 °C. The flowthrough was collected to monitor the dissociation of pre-bound proteins, and the protein complexes were washed, eluted and examined as described above.

To analyze the competition observed is in a passive or active manner, we pre-assembled the core CTC and RSAD2 complex as described above and monitored the dissociation of RSAD2 during an extensive washing process with buffer-only and buffer containing 5 µM GB1-tagged CDKAL1 proteins, respectively. In this process, the column flow rate was kept at approximately 0.08 ml/min, and the total washing volume was extended to 50 column volumes (CV, at approximately 10 ml). 1% (v/v) of the flowthrough was collected at the indicated points as samples and analyzed by SDS-PAGE gels.

### Isothermal titration calorimetry (ITC)

All protein samples used in ITC underwent buffer exchange to 1×PBS via size exclusion chromatography as the last step of purification. ITC measurements were performed at 20 °C with a MicroCal PEAQ-ITC or PEAQ-ITC Automated calorimeter (Malvern Panalytical). The typical titration was carried out with 20 µM CTC or its components in the cell as titrate and 200 µM indicated protein or its variants in the syringe as titrant. Some variants were loaded into the syringe at 300–600 µM due to the weak binding. Data were analyzed by MicroCal PEAQ-ITC Analysis Software and fitted to the one set of sites binding model.

### Crystallization, data collection and structure determination

To crystalize the protein complex of heterodimeric CIAO1-CIAO2B and CDKAL1-N or any of its fragments containing determinant regions, we generated a chimeric CIAO2B by replacing a native sequence of 'EIENI' at the N-terminal flexible region of *dm*CIAO2B with the *h*CDKAL1 pentapeptide of 'DIEDI', largely due to their high sequence similarity. The complex of *dm*CIAO1-CIAO2B and *h*CDKAL1 pentapeptide was produced on the basis of this chimeric CIAO2B and applied to crystallization screening. The best diffracted crystals were obtained at 4 °C by the hanging-drop vapor diffusion method, using protein sample at 15 mg/ml mixed with an equal volume of reservoir solution containing 0.2 M Sodium phosphate dibasic dihydrate, and 20% (w/v) PEG 3350, pH 9.1. Crystals appeared within 1–2 days and matured to full size in approximately a week. Crystals were cryoprotected by reservoir solution supplemented with 20% (v/v) glycerol in harvest, and flash frozen in liquid nitrogen for data collection. All datasets were collected at 100 K with a wavelength of 1.000 Å at the BL8.2.2 beamline at the Advanced Light Source of the Lawrence Berkeley National Laboratory. Diffraction data were indexed, integrated, and scaled with the HKL2000 package (Otwinowski and Minor, 1997). The structure was determined by molecular replacement using

Phaser (McCoy et al, 2007) in the CCP4 suite (Agirre et al, 2023) with the structure of *dm*CIAO1-CIAO2B (PDB: 6TBN) as a searching model. Model building and refinement were performed using Coot (Emsley et al, 2010) and PHENIX (Liebschner et al, 2019). Ramachandran plot analysis of the final model showed that 94.35% and 5.65% of the residues are in favored and allowed regions, respectively. Complete data collection and refinement statistics are summarized in Table EV1. All structure figures were rendered in PyMOL.

## Iron and sulfide content measurements and UV/Vis spectroscopy

Iron content of purified FLAG-tagged WT Fe-S proteins and their variants were measured by ferrozine based colorimetric assay (MAK025, Sigma-Aldrich). 100 µg of protein was applied, and the iron released by the addition of an acidic buffer was reduced to measure both $Fe^{2+}$ and $Fe^{3+}$ in the samples. The total iron reacted with a chromagen and the released product was determined colorimetrically at 593 nm, which was proportional to the iron amount. A standard curve covering 0.2 nmole to 10 nmole/well iron was generated using iron standard supplied in the kit. All colorimetric readings from the reactions in transparent 96-well plates were taken by Synergy HTX plate reader (BioTek).

Quantitative determination of labile sulfide in the purified FLAG-tagged WT CDKAL1 and its pentapeptide-deficient variant were conducted according to the methylene blue method previously described (Rabinowitz, 1978). Briefly, 150 µg of protein was incubated 30 min with an alkaline zinc reagent formed by 86.7 µl 1% zinc acetate and 3.3 µl 12% NaOH. 16.7 µl 0.1% diamine reagent (N,N-Dimethyl-p-phenylenediamine dihydrochloride) was applied to dissolve zinc precipitates and the methylene blue was formed by the addition of 3.3 µl 23 mM $FeCl_3$. The reaction mixture was diluted with water to 200 µl in transparent 96-well plates for colorimetric detection at 670 nm. A standard curve covering 0.5 nmole to 10 nmole/well sulfide was generated using freshly prepared $Na_2S$. All colorimetric readings were taken by Synergy HTX plate reader (BioTek) after precipitates removed.

The UV/Vis spectra of purified FLAG-tagged WT Fe-S proteins and their variants were recorded with sealed micro cuvettes using NanoDrop One$^C$ spectrophotometer (Thermo Fisher Scientific). The protein samples were at the concentration of 2 mg/ml in the buffer containing 20 mM Tris-HCl pH 7.5, 150 mM NaCl, 0.2 mg/ml 3 × FLAG peptide, 5 mM DTT.

## Helicase assay

A 5'-cy5-labeled 42mer oligonucleotide (GACGCTGCCGAATTCTAC-CAGTGCCTTGCTAGGACATCTTTG) was annealed with 5' (CAA AGATGTCCTAGCAAGGCTTTTTTTTTTTTTTTTTTTTTT) 3' to make Y-structure substrate. DNA unwinding reaction mixtures contained 25 mM Tris-HCl pH 7.5, 1 mM DTT, 100 µg/ml BSA, 5 mM $MgCl_2$, 5 mM ATP, 60 mM NaCl, 1 nM Y-structure substrate and the indicated concentrations of FANCJ helicase. The reactions were initiated by the addition of FANCJ and then incubated at 37 °C for 30 min. Reactions were quenched in the presence of 0.25 mg/ml Proteinase K, 0.1% SDS and 5-fold excess of unlabeled oligonucleotide, same sequence as the labeled strand at 37 °C for 10 min. The products were resolved on nondenaturing 8% native polyacrylamide gels by

adding 6×gel loading buffer (60% glycerol, 120 mM Tris-HCl, pH 7.4, 3 mM EDTA, 0.025% orange G).

## Glycosylase assay

A 5'-cy5-labeled 39mer oligonucleotide (TGAGACTGGCCAGC-TAACTGAACTGATCATGCCTAGCGT) was hybridized with its complementary strand (ACGCTAGGCATGATCAGTXCAGT-TAGCTGGCCAGTCTCA, where X = 8-hydroxyguanine (GO)) to make the dsDNA substrate containing a A:GO mismatch. The adenine DNA glycosylase activity reaction mixtures contained 25 mM Tris-HCl pH 7.5, 1 mM DTT, 100 µg/ml BSA, 50 µM $ZnCl_2$ and 60 mM NaCl, 0.3 nM substrate and indicated concentrations of MUTYH. The reactions were initiated with the addition of MUTYH by incubating at 37 °C for 30 min. The reaction was stopped by adding 2.5 µl 1 M NaOH, heating at 90 °C for 10 min. The samples were mixed with final 27% of formamide and 3.5 µl 6×gel loading buffer, heated at 90 °C for 10 min, and electrophoresed on a 18% denaturing (7 M Urea) PAGE gel.

## Nuclease assay

Nuclease reaction mixtures contained 25 mM Tris-HCl pH 7.5, 1 mM DTT, 100 µg/ml BSA, 5 mM $MgCl_2$, 5 mM ATP, 45 mM NaCl, 1 nM 5'-cy5-labeled 42mer substrate (GACGCTGCC-GAATTCTACCAGTGCCTTGCTAGGACATCTTTG) and the indicated concentrations of DNA2. The reactions were initiated by the addition of DNA2 and then incubated at 37 °C for 30 min. Reactions were quenched in the presence of 0.25 mg/ml Proteinase K, 0.1% SDS by incubating at 37 °C for 10 min. The products were resolved on nondenaturing 10% native polyacrylamide gels adding 6×gel loading buffer.

## Mammalian cell culture

HEK293T, HEK293FT and HeLa cell lines were purchased from MeilunBio and routinely monitored for *Mycoplasma* contamination using the Universal Mycoplasma Detection Kit (Yeasen Biotechnology). Cells were maintained in Dulbecco's modified Eagle's medium (DMEM) (Gibco) supplemented with 10% fetal bovine serum (FBS) (Sigma-Aldrich) and 1% Penicillin-Streptomycin (Gibco) at 37 °C and 5% $CO_2$ in a humidified atmosphere.

## Immunoprecipitation and immunoblotting

HEK293T cells were transiently transfected with the indicated plasmids using Lipofectamine 3000 reagent (Invitrogen) for 48 h. Cells were harvested and lysed in lysis buffer (Beyotime) supplemented with protease inhibitor cocktail (Sigma-Aldrich) on ice. The insoluble fraction was removed by high-speed centrifugation for 20 min at 4 °C. Immunoprecipitations of FLAG-tagged proteins were performed using pre-equilibrated anti-FLAG Magnetic Agarose (Thermo Scientific) for 2 h at 4 °C. The beads were then extensively washed by lysis buffer. Whole cell lysates and the immunoprecipitates were separated by SDS-PAGE and transferred to PVDF membranes (Millipore) for Western blotting at 4 °C. Transfer efficiency was checked by Ponceau S (Beyotime) staining. Membranes were then blocked in 5% milk/PBST for 1 h at room

temperature and incubated with the indicated primary antibodies at 4 °C overnight.

The detection of proteins was accomplished using the appropriate secondary antibodies conjugated to horseradish peroxidase in PBST. Western blots were developed using SuperSignal West Pico PLUS Chemiluminescent Substrate (Thermo Scientific) on Amersham ImageQuant 800 (Cytiva).

## CRISPR-Cas9 genome editing

To generate HeLa FANCJ KO cell lines, two sgRNAs (sgRNA-1: GTCATCGAATACCATTAAGA, sgRNA-2: CCGGAGGACGG-CATATTCAG) targeting Exon 6 on FANCJ were designed and cloned into the pSpCas9(BB)-2A-GFP (PX458) vector, respectively. Two plasmids were co-transfected into HeLa cells using Lipofecta-mine 3000. GFP-positive single-cell clones were sorted into 96-well plates 48 h post-transfection using fluorescence-activated cell sorting (FACSAria III, BD) and grow. Genotyping PCRs were conducted with extracted genomic DNA as templates and the following primers: F-5'-GTAGTTTTTTCTTTAATGGCAAGTTT-TAGA and R-5'-GTTAATTTGATTTTCCGAAGTTGATTAT-CAC. PCR products from the promising clones were purified and sequenced to determine the positive clones with gene disruption. The positive clones were further validated by western blotting.

## Lentivirus production and stable cell line generation

HEK293FT cells were transfected with pLVML-3×FLAG vectors containing FANCJ WT or its variants, in combination with pMD2.G and psPAX2 at a ratio of 4:3:1 using polyethylenimine (PEI, FUSHENBIO). Lentiviral particles in media were harvested 48 h after transfection and filtered by 0.45 μm filter (Millipore). HeLa FANCJ KO cells were then infected with the lentiviral particles for 24 h. Following transduction, cells were selected by 1–2 μg/ml puromycin for a week and pool of puromycin-resistant cells were confirmed by immunoblotting.

## Homologous recombination assay

Cell lines of HeLa, HeLa-FANCJ$^{KO}$, and HeLa-FANCJ$^{KO}$-FANCJ$^{EV/WT/M1/M2}$ were seeded in 6-well plates at $2.5 \times 10^5$ cells per well. The vectors of pDRGFP and I-SceI (pCBASceI) purchased from MiaoLingPlasmid were co-transfected into the cells using Lipofectamine 3000 the second day. Three days after transfection, the HR proficiency was determined by counting the fraction of GFP-positive cells using a BD FACS Celesta. The results were derived from three biological replicates.

## Clonogenic survival assay

Cell lines of HeLa, HeLa-FANCJ$^{KO}$, HeLa-FANCJ$^{KO}$-FANCJ$^{EV/WT/M1/M2}$ were seeded in 12-well plates at 400 cells per well and treated with indicated amount of mitomycin C (MMC, Selleck) in standard growth medium for 9–10 days. Cells were then washed with PBS, fixed with methanol and stained with 0.5% crystal violet in methanol. The colony number was analyzed using ColonyArea ImageJ plugin and the clonogenic survival curve was determined by dividing the number of colonies on each treated plate by the number of colonies on the untreated plate.

## Affinity purification mass spectrometry (AP-MS) and data analysis

The human CTC WT and its two pocket-deficient variants (CIAO1$^{R125A}$-CIAO2B-MMS19 and CIAO1-CIAO2B$^{R141A}$-MMS19) were assembled and purified in vitro on the basis of N-terminal 3×StrepII-tagged CIAO1. These three complexes were immobilized on the pre-equilibrated Streptavidin Magnetic Beads (Thermo Scientific) as baits. The lysate from DFO-treated HEK293T cells (treatment of 100 μM DFO for 12 h) was precleaned by Streptavidin beads to remove biotin-binding and non-specific binding proteins. 20 μl bait-bound beads were incubated with 5–10 mg total proteins from pre-treated cell lysate at 4 °C for one hour. After extensive wash by lysis buffer, the beads were resolved by SDS-PAGE and analyzed by silver staining (Pierce Silver Stain Kit, Thermo Scientific). To prepare samples for mass spectrometry, the SDS-PAGE gel covering all protein bands of interest was excised. The following in-gel trypsin digestion, peptide extraction, MS data acquisition and preliminary analysis were performed by Omicsolu-tion Ltd. All proteomic data were searched against human proteome (Uniprot reviewed sequences in 2024). Detected peptides and proteins were filtered to 1% false-discovery rate (FDR), and the proteins containing at least one unique peptide were identified. The results were derived from two technical replicates and shown in Dataset EV1.

Average spectral counts of known human Fe-S proteins captured in the AP-MS were visualized by dot plot generated in ProHits-viz program (Knight et al, 2017). A two-step filtering strategy was applied to explore the unrecognized human Fe-S proteins. In the first step, we extracted all the CTC-interactors that were considered to be pocket-dependent with the ratio of average spectral counts of CTC-WT to that of both pocket-deficient variants ≥5 and the average spectral counts of CTC-WT ≥ 5. In the second step, among all the pocket-dependent CTC-interactors, proteins localized to the cytoplasm, or the nucleus were chosen to be the hits of interest with high confidence. Proteins as high-confidence hits of interest were visualized both by dot plot generated in ProHits-viz program and with Cytoscape (Shannon et al, 2003), either individually or grouped as protein complexes and biological processes according to their reported functions.

## Antibodies

Antibodies used for immunoblotting were anti-CIAO1 (1:1000, Cell Signaling Technology #81376), anti-MMS19 (1:500, Santa Cruz Biotechnology sc-390658), anti-CIAO2B (1:1000, Cell Signaling Technology #86302), anti-β-Tubulin (1:1000, Cell Signaling Technology #15115), anti-DYKDDDDK Tag (1:1000, Cell Signal-ing Technology #14793), anti-FANCJ (1:1000, Cell Signaling Technology #4578), anti-rabbit IgG, HRP-linked antibody (1:2000, Cell Signaling Technology #7074), and anti-mouse IgG, HRP-linked antibody (1:2000, Cell Signaling Technology #7076).

## Statistical analysis

All statistical analysis was performed on the basis of three technical or biological replicates unless specially indicated, and analyzed with GraphPad Prism software. Data are represented as mean ± standard deviation (SD) or mean ± standard error of the mean (SEM), as

indicated in the figure legends. Statistical significance was analyzed using two-way ANOVA or two-tailed unpaired t-test.

## Data availability

Atomic coordinates of the *dm*CIAO1-CIAO2B-*h*CDKAL1 penta-peptide complex have been deposited to the Protein Data Bank (PDB) (www.rcsb.org) with the accession number PDB: 9UKY (https://doi.org/10.2210/pdb9uky/pdb).

The source data of this paper are collected in the following database record: biostudies:S-SCDT-10_1038-S44318-025-00676-x.

## Peer review information

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

## Acknowledgements

We thank Dr. Ning Zheng for help on diffraction data collection, and insightful comments and discussions on the manuscript, Keng Chen and Yijing Lu for technical support in flow cytometry, Jiying Hu for assistance in ITC experiments, and the members of Wang lab for discussion and help. The authors thank beamline 8.2.2 and the staff of the Advanced Light Source, which is a DOE Office of Science User Facility under Contract No. DE-AC02-05CH11231, supported in part by the ALS-ENABLE program funded by the National Institutes of Health, National Institute of General Medical Sciences, grant P30 GM124169-01. This work was supported by grants from the China Postdoctoral Science Foundation (2024M752148 to YXH), the National Natural Science Foundation of China (22577079 to HW), and the Shenzhen Bay Laboratory Start-up Funds (21320081 to HW).

## Author contributions

**Wenjie Ren**: Data curation; Formal analysis; Investigation; Writing—review and editing. **Yuxin Huang**: Data curation; Formal analysis; Funding acquisition; Investigation; Methodology; Writing—review and editing. **Min Hu**: Data curation; Investigation. **Yanyang Yang**: Data curation; Formal analysis; Investigation. **Wen Yang**: Conceptualization; Formal analysis; Investigation; Methodology; Project administration; Writing—review and editing. **Hui Wang**: Conceptualization; Supervision; Funding acquisition; Methodology; Writing—original draft; Writing—review and editing.

Source data underlying figure panels in this paper may have individual authorship assigned. Where available, figure panel/source data authorship is listed in the following database record: biostudies:S-SCDT-10_1038-S44318-025-00676-x.

## Disclosure and competing interests statement

The authors declare no competing interests.

# Expanded View Figures

**Figure EV1.  Probing the interaction between the CTC and CDKAL1.**

(**A**) Schematic drawing of the CTC complete complex and the core CTC. (**B**) Proteins utilized as input in GST pull-down assay shown in Fig. 1A. (**C**) Proteins utilized as input in GST pull-down assay shown in Fig. 1C. (**D**) ITC data utilized to generate the quantitative binding affinities shown in Fig. 1B. The proteins and protein complexes used in each titration are listed above the curves. (**E**) ITC data utilized to generate the quantitative binding affinities shown in Fig. 1E. The protein fragments and protein complexes used in each titration are listed above the curves.

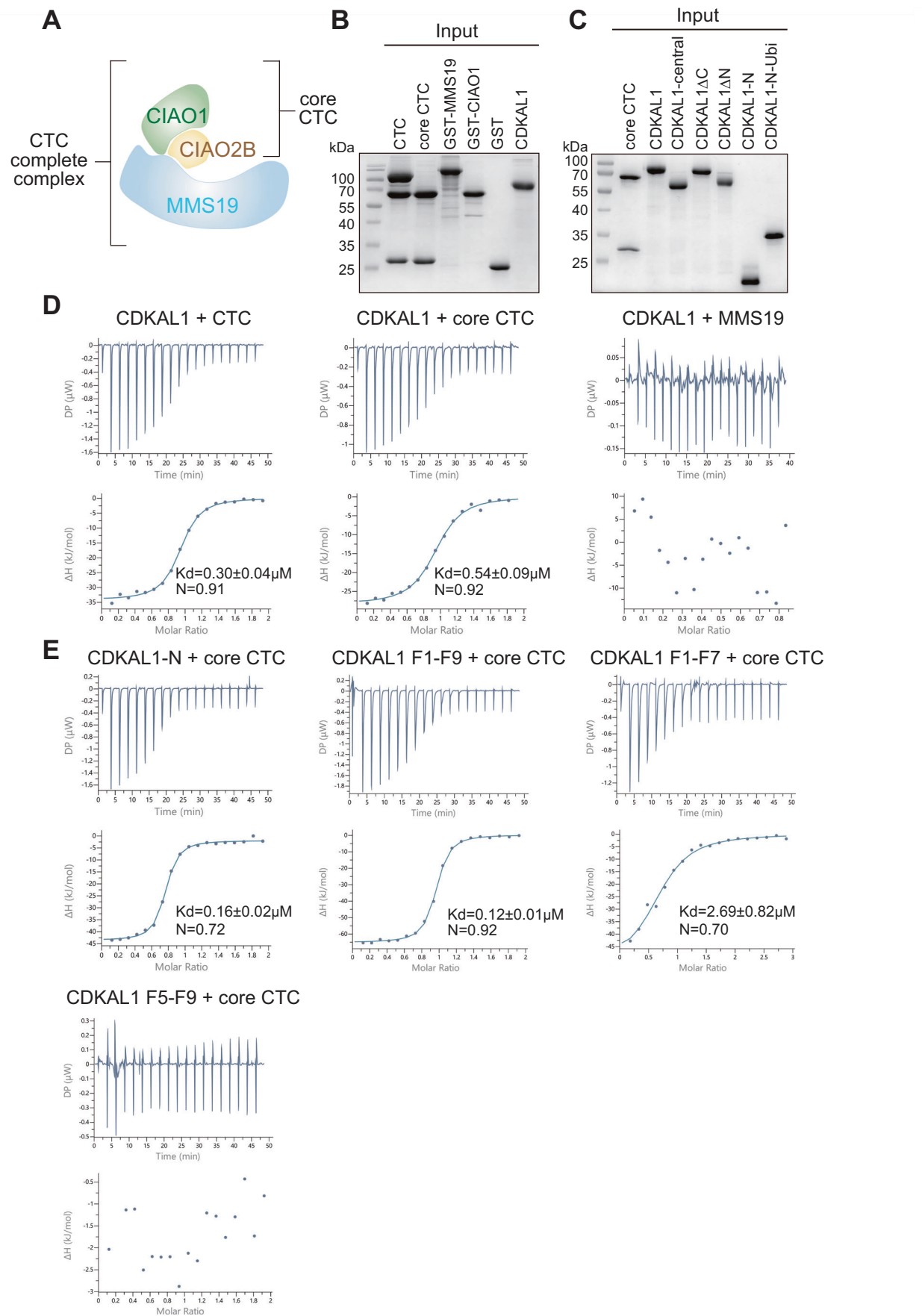

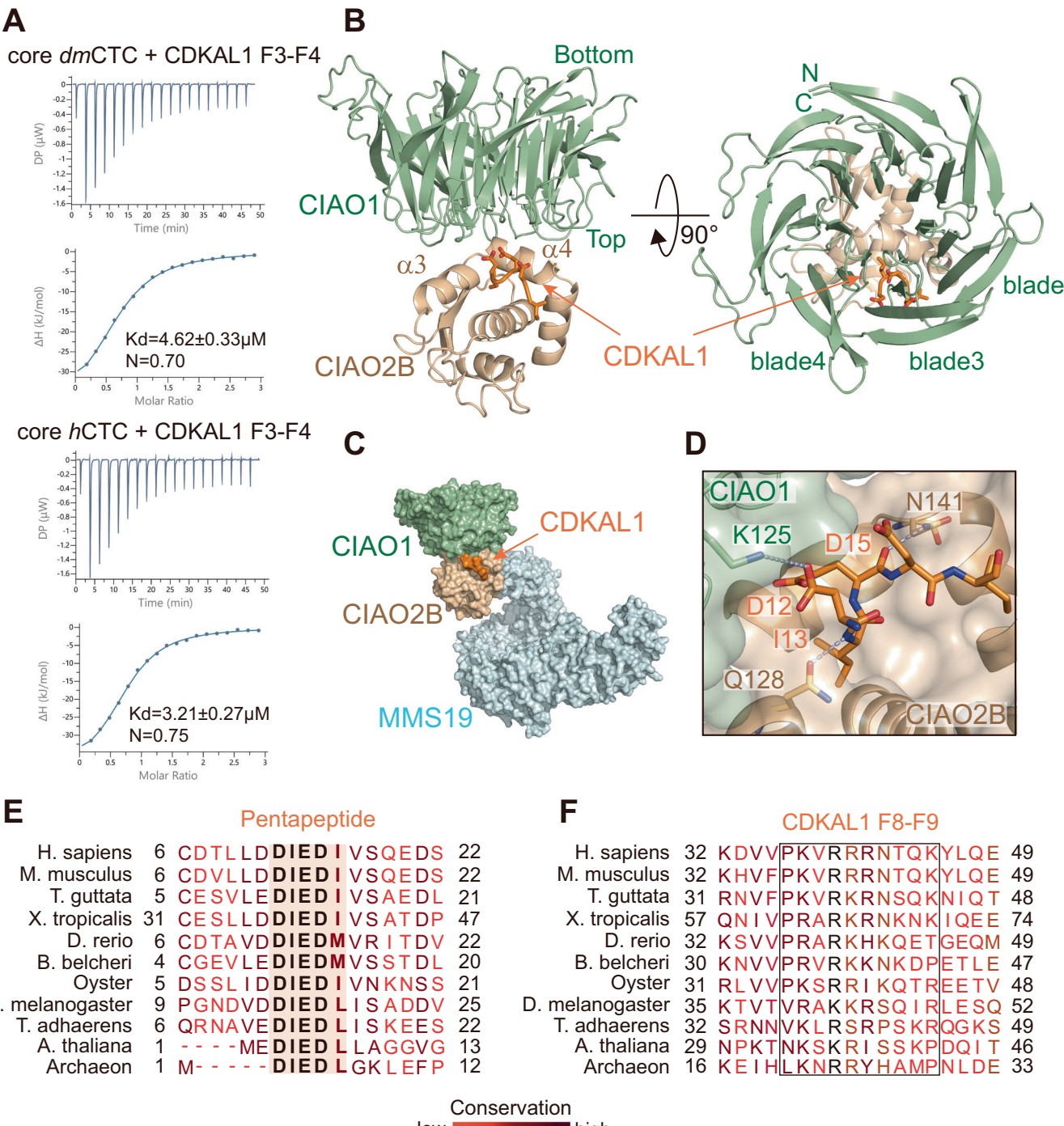

◀ **Figure EV2.    CDKAL1 pentapeptide mediating the CDKAL1-CTC interface.**

(A) ITC data showing the quantitative binding affinities of CDKAL1 fragment F3–F4 to the core CTC in both *Drosophila* and human versions. (B) Two orthogonal views of the complex comprising *dm*CIAO1-CIAO2B and *h*CDKAL1 pentapeptide shown in ribbon diagram with CIAO1 in sage green, CIAO2B in wheat, and CDKAL1 pentapeptide in orange, respectively. The side chains of pentapeptide are shown in sticks. The N and C termini and the top and bottom sides of CIAO1 are labeled accordingly in green. Select secondary structure/structural unit of different proteins are numbered and labeled in corresponding colors. (C) Structural model of CDKAL1 pentapeptide in complex with CTC complete complex composed of CIAO1, CIAO2B, and MMS19 generated by superposition of the structures of *dm*CIAO1-CIAO2B-*h*CDKAL1 pentapeptide and *dm*CIAO1-CIAO2B-*m*MMS19 (PDB: 6TC0). The model is shown in surface with the same color scheme as shown in (B), and MMS19 is shown in the color of light blue. (D) Close-up view of the interface between CDKAL1 pentapeptide and *dm*CIAO1-CIAO2B complex featuring supportive contacts targeting the backbone and side chain of CDKAL1 pentapeptide. CIAO1 and CIAO2B are shown in surface and ribbon with the same color scheme as shown in (B), and CDKAL1 pentapeptide is shown in orange sticks. Select interface residues are shown in sticks. Dashed lines in violet blue represent hydrogen bonds and polar interactions. (E, F) Sequence alignments of the regions enclosing the pentapeptide and the fragment F8–F9 in CDKAL1 orthologs from *Homo sapiens*, *Mus musculus*, *Taeniopygia guttata*, *Xenopus tropicalis*, *Danio rerio*, *Branchiostoma belcheri*, *Oyster*, *Drosophila melanogaster*, *Trichoplax adhaerens*, *Arabidopsis thaliana* and *Archaeon*. Residues are colored from red to black according to their conservations from low to high. The sequences of pentapeptide and fragment F8–F9 are highlighted by orange shade and with a frame in black, respectively.

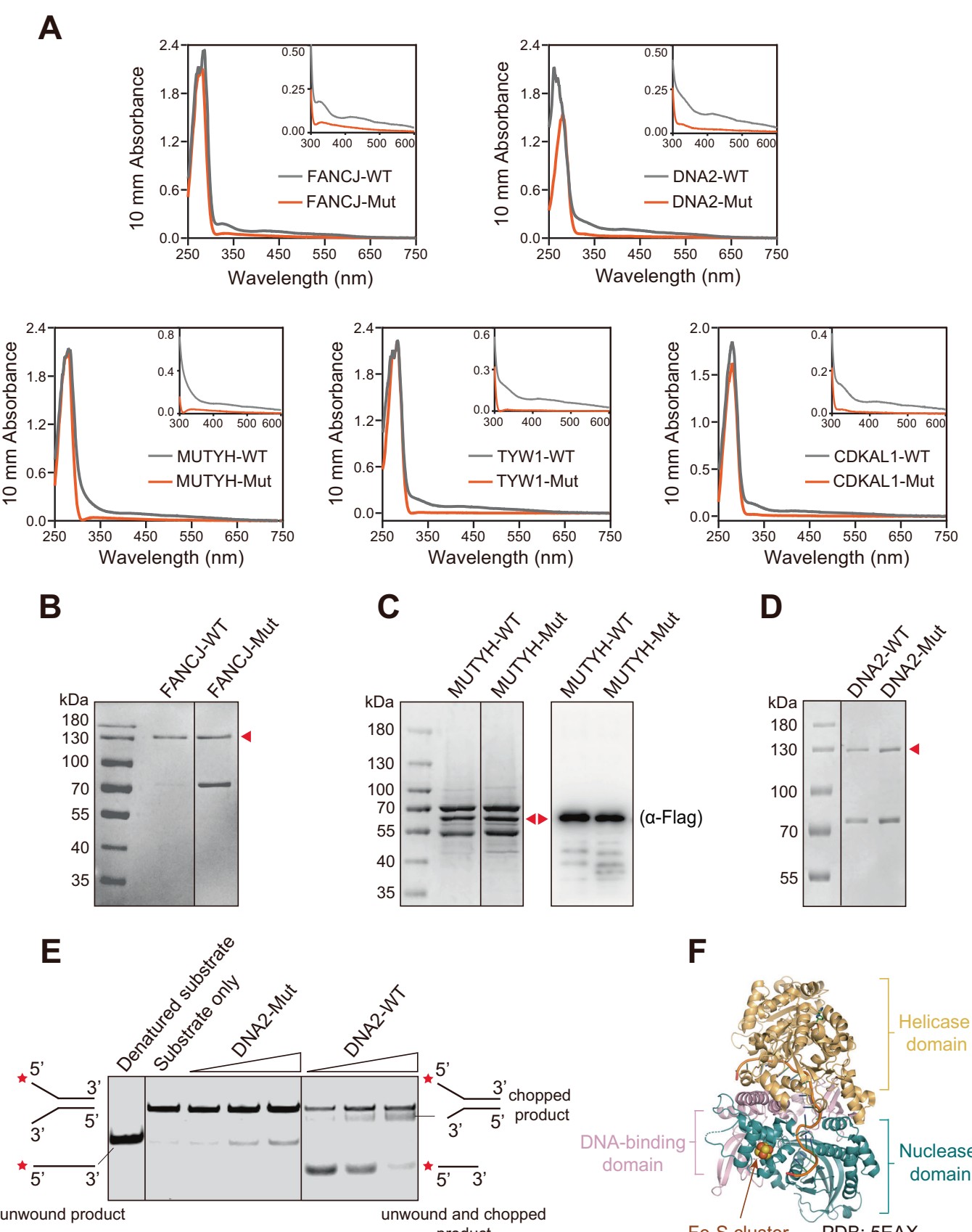

Figure EV3.   Biochemical analyses of Fe-S proteins in vitro.

(A) UV/Vis spectra (250–750 nm range) of purified FLAG-tagged WT Fe-S proteins and their variants. The inset zoom-in views were displayed from 300 to 600 nm for improved clarity. (B–D) SDS-PAGE of purified FLAG-tagged proteins of (B) FANCJ[1-1100aa], (C) MUTYH[FL], and (D) DNA2[FL] in forms of WT and pentapeptide-deficient mutant. MUTYH[FL] WT and mutant are also detected by immunoblotting via α-FLAG antibody. The red arrows indicate the target proteins. (E) DNA helicase-nuclease assay with [cy5]Y-substrate comparing 10 nM, 20 nM, 40 nM FLAG-DNA2 WT and 10 nM, 20 nM, 40 nM FLAG-DNA2 mutant. Heat-denatured substrate indicating the unwound product and substrate only without proteins are shown as controls. (F) Structure of DNA2 in complex with ssDNA (PDB: 5EAX) shown in ribbon diagram with its nuclease domain in teal, helicase domain in yellow, DNA-binding domain in pink, and ssDNA in orange, respectively. The ATP molecule is shown in sticks, and the Fe-S cluster is shown in spheres.

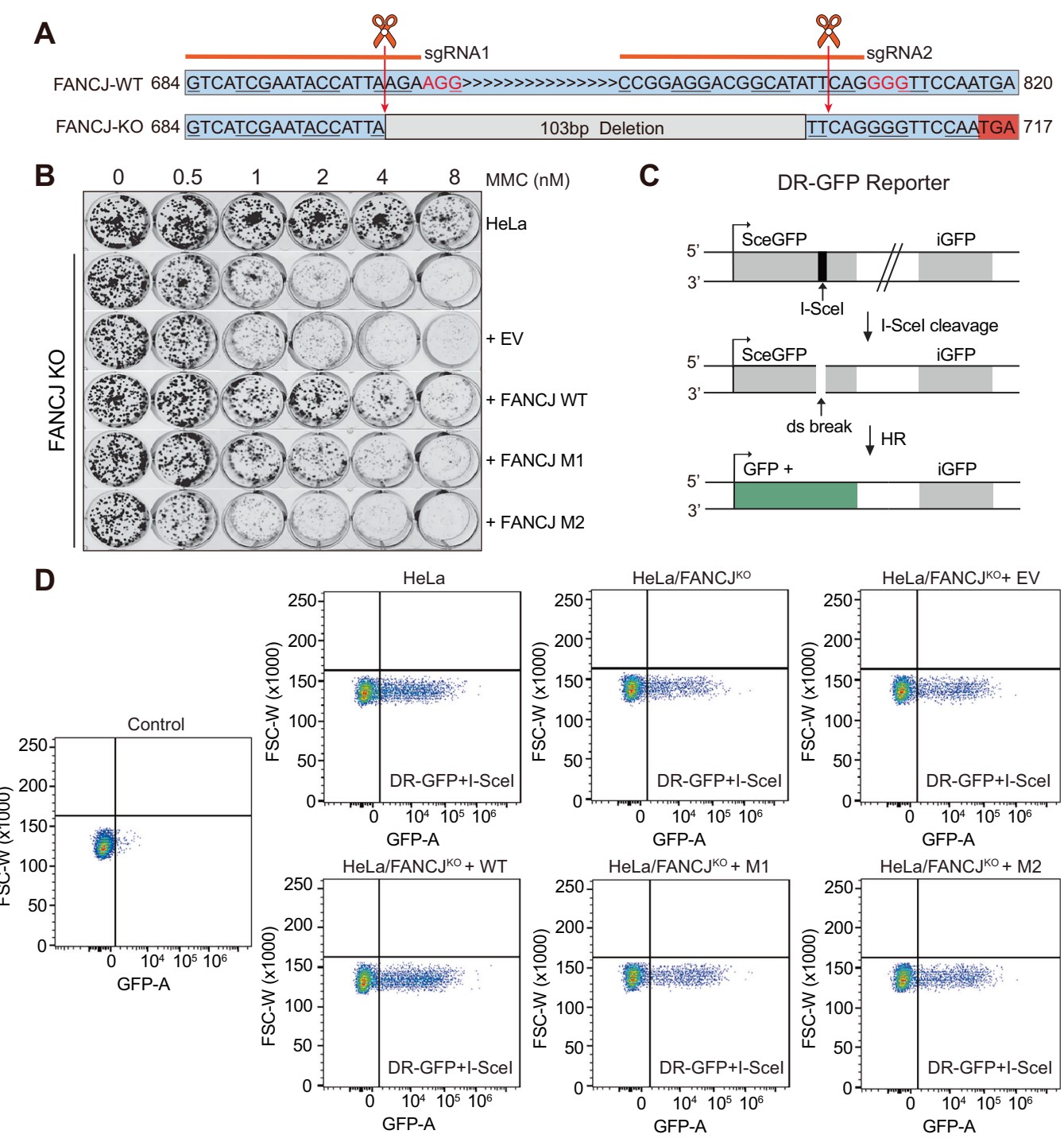

**Figure EV4. Functional assays of FANCJ in cells.**

(**A**) A design for CRISPR/Cas9 mediated FANCJ KO in HeLa cells. sgRNA1 and sgRNA2 are labeled and the PAM motifs are highlighted in red. The stop codon created in the editing product is indicated by rectangular in red. (**B**) Representative images of the clonogenic survival assay under MMC treatment using HeLa and HeLa-derived cell lines in FANCJ deletion/complementation system. EV: empty vector. (**C**) Schematic diagram of the HR reporter assay. (**D**) Representative HR assay FACS plots for control and I-SceI/DR-GFP-transfected HeLa and HeLa-derived cell lines in FANCJ deletion/complementation system. EV: empty vector.

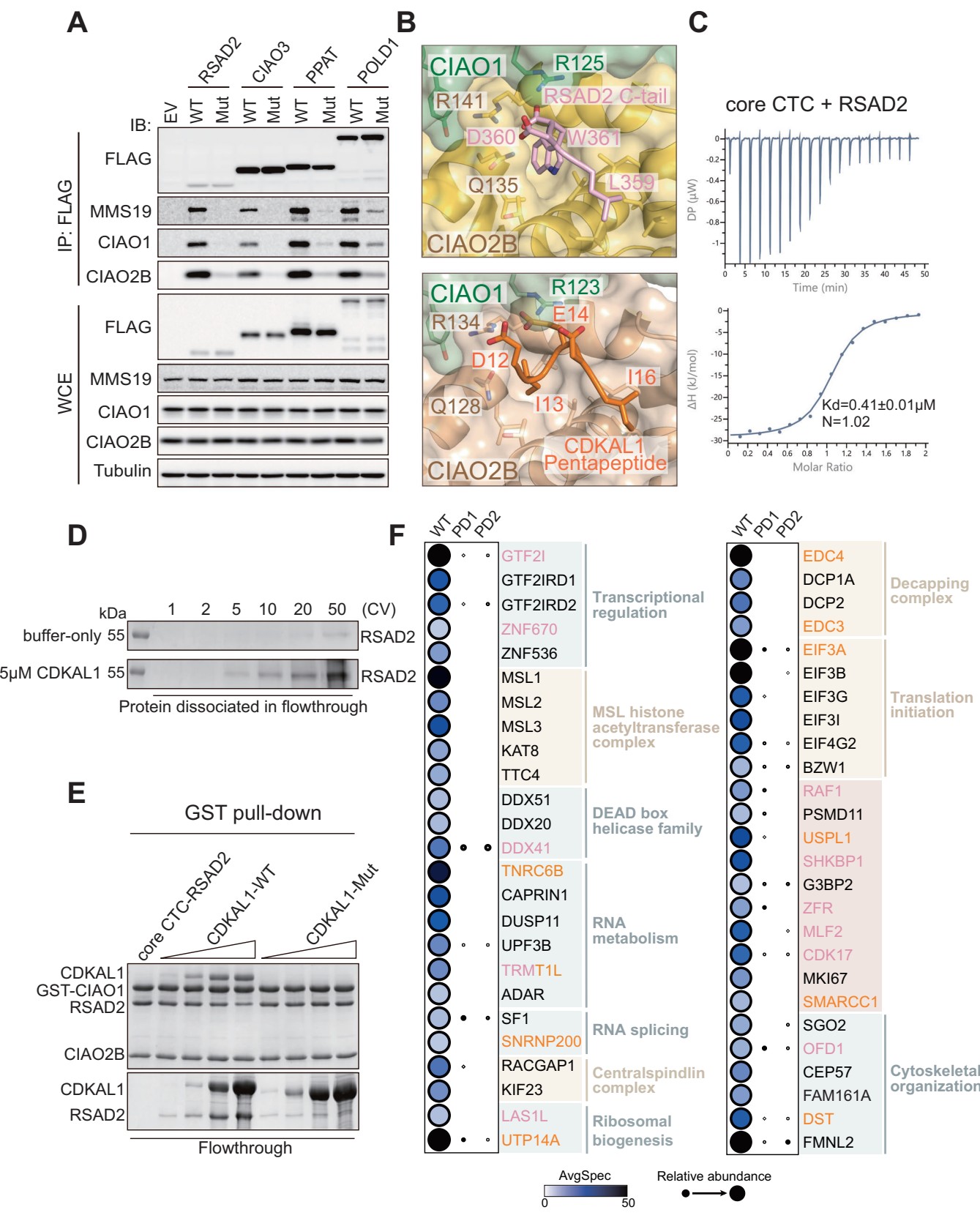

◀ **Figure EV5. The CTC client recruitment pocket and differential AP-MS.**

(A) HEK293T cells were transiently transfected with FLAG-tagged indicated Fe-S proteins in forms of either WT or C-tail motif-deficient mutant (specifically, an alanine substitution for the C-tail tryptophan residue). Whole cell extracts (WCE) were subjected to immunoprecipitations with anti-FLAG beads and immunoblotting against the antibodies of FLAG, MMS19, CIAO1 and CIAO2B. EV: empty vector. (B) Close-up views of the interfaces between a shared pocket in CTC and both RSAD2 C-tail (upper panel) and CDKAL1 pentapeptide (lower panel) generated on the basis of the structural model shown in Fig. 6A. CIAO1 and CIAO2B are shown in surface and ribbon with CIAO1 in shades of green and CIAO2B in shades of yellow. RSAD2 C-tail is shown in pink ribbon with side chains in sticks, while CDKAL1 pentapeptide is shown in orange ribbon with side chains in sticks. Select interface residues are labeled and shown in sticks. The amino acid numbering of CIAO1-CIAO2B complex in the lower panel is based on the *Drosophila* version resolved in our structure, while all the rest molecules are numbered as human versions. (C) ITC data showing the quantitative binding affinity of RSAD2 to the core CTC. (D) Client dissociation or displacement from a pre-assembled core CTC-RSAD2 complex during an extensive washing process with buffer-only and buffer containing 5 μM GB1-tagged CDKAL1 proteins, respectively. The loaded sample represents 1% (v/v) of the flowthrough collected at the indicated points (1, 2, 5, 10, 20, 50 CV). (E) In vitro competition assays demonstrating the competitive abilities of CDKAL1 WT and its pentapeptide-deficient mutant against RSAD2 for CTC binding. (F) Dot plot generated in ProHits-viz program visualizing the average spectral counts of proteins as high-confidence hits of interest captured in the AP-MS by using the CTC WT and pocket-deficient variants PD1 and PD2 as baits. Fill color and size of dots represent AvgSpec and Relative abundance, respectively. The fill color represents the average spectral count of each protein, with darker shading indicating higher counts (maximum capped at 50; values above 50 are shown in black). The dot size reflects the relative abundance of each protein across the three datasets. An explanation of the filtering strategy is detailed in the methods section. Proteins are shown either individually or grouped as protein complexes and biological processes according to their reported functions. The Pentapeptide-containing and C-tail-containing proteins are colored in orange and pink, respectively. The protein complexes and biological processes are labeled and colored accordingly. PD1: Pocket-deficient 1, containing CIAO1-R125A, and PD2: Pocket-deficient 2, containing CIAO2B-R141A.

