## [Peer Review File · The EMBO Journal]

Client Recruitment Mechanism of the Cytosolic Fe-S Cluster Assembly Targeting Complex

Wenjie Ren, Yuxin Huang, Min Hu, Yanyang Yang, Wen Yang, and Hui Wang

Corresponding author(s): Hui Wang (hwang@szbl.ac.cn) , Wen Yang (yangwen@szbl.ac.cn)

Review Timeline:

Submission Date:	4th Jun 25
Editorial Decision:	27th Jun 25
Revision Received:	22nd Sep 25
Editorial Decision:	20th Oct 25
Revision Received:	8th Dec 25
Accepted:	12th Dec 25

Editor: Hartmut Vodermaier

Transaction Report:

Dr. Hui Wang
Shenzhen Bay Laboratory
Greater Bay Biomedical InnoCenter
Shenzhen, Guangdong 518132
China

27th Jun 2025

Re: EMBOJ-2025-121548
Client Recruitment Mechanism of the Cytosolic Fe-S Cluster Assembly Targeting Complex

Dear Dr. Wang,

Thank you for submitting your study on client protein recruitment to the cytosolic Fe-S cluster assembly targeting complex for our editorial consideration. We have now received detailed comments from two expert reviewers, copied below for your information. Since both referees find the work significant and generally well-performed, we would be happy to consider an adequately revised version further for publication in The EMBO Journal.

As you will see, the majority of issues raised in the two reports are related to various presentational aspects; however referee 2 also brings up a number of data-related questions, especially in their general comments, which would need to be addressed. Since it is our policy to allow only a single round of major revision, please do not hesitate to contact me in case you would like to clarify/discuss any of the referees' points or plans for answering. We would also be open to extending the revision deadline if that should be helpful. Our scooping protection policy means that competing manuscripts published while your work is under revision will not have a negative effect on our final decision.

Detailed information on preparing, formatting and uploading a revised manuscript can be found below and in our Guide to Authors, and adhering to them as closely as possible shall greatly facilitate editorial processing upon resubmission. Thank you again for the opportunity to consider this work for The EMBO Journal, and I look forward to your revision in due time.

Yours sincerely,

Hartmut Vodermaier

*** PLEASE NOTE: All revised manuscripts are subject to initial checks for completeness and adherence to our formatting guidelines. Revisions may be returned to the authors and delayed in their editorial re-evaluation if they fail to comply to the following requirements (see also our Guide to Authors for further information):

- size of the scale bars that are mandatory for all micrograph panels
- the statistical test used to generate error bars and P-values
- the type of error bars (e.g., S.E.M., S.D.)
- the number (n) and nature (biological or technical replicate) of independent experiments underlying each data point
- Figures may not include error bars for experiments with $n < 3$; scatter plots showing individual data points should be used instead.

4) Each main and each Expanded View (EV) figure should be uploaded as individual production-quality files (preferably in .eps, .tif, .jpg formats). For suggestions on figure preparation/layout, please refer to our Figure Preparation Guidelines:

9) To facilitate reproducibility and cross-laboratory adoption of methodologies, please structure the Materials & Methods section as outlined in our guide to authors, including a completed Reagents and Tools Table that can be downloaded from our author guidelines as well (<https://www.embopress.org/page/journal/14602075/authorguide#structuredmethods>).

10) Digital image enhancement is acceptable practice, as long as it accurately represents the original data and conforms to community standards. If a figure has been subjected to significant electronic manipulation, this must be clearly noted in the figure legend and/or the 'Materials and Methods' section. The editors reserve the right to request original versions of figures and the original images that were used to assemble the figure. Finally, we generally encourage uploading of numerical as well as gel/blot image source data; for details see: embopress.org/page/journal/14602075/authorguide#sourcedata

In the interest of ensuring the conceptual advance provided by the work, we recommend submitting a revision within 3 months (25th Sep 2025). Please discuss the revision progress ahead of this time with the editor if you require more time to complete the revisions. Use the link below to submit your revision:

Link Not Available

Referee #1:

In this paper, the authors discover that a distinctive pentapeptide motif in proteins that acquire FeS cofactors in the cytosolic/nuclear compartment binds to a pocket in CIAO1 and its interface with CIA2. They use CDKAL1 for their experiments, as a representative, and they use sequence homology, alanine scanning, and affinity co-immunoprecipitations to prove their point. Then they examine other known FeS proteins in the cytosolic nuclear compartment and discover that the pentapeptide motif is present in many proteins of RNA and DNA metabolism. They model the pocket in the complex of CIAO1 and CIA2, and discover that proteins that acquire their FeS using a terminal W/F motif bind in the same pocket. Their work generates important insights into the molecular interactions that promote acquisition of critical FeS cofactors in many important proteins involved in genome maintenance, and their work also allows prospective identification of FeS candidate proteins that were not previously recognized. It is an excellent body of work that will energize FeS studies.

There are several aspects of the paper that need correction or clarification:

1. In the abstract, they state that maturation of most eukaryotic Fe-S proteins requires the cytosolic Fe-S... On this point, it would be better to state something like- Most eukaryotic Fe-S proteins acquire their critical Fe-S cofactor by interacting with the cytosolic Fe-S cluster assembly targeting complex... Use of the word maturation is not universally accepted and description can be more precise, particularly in the abstract, which most people read before they commit to reading the paper.
2. They do not correctly represent results from papers written by the Rouault group. They need to read these papers and correctly assert that the Rouault group has found that the initial Fe-S assembly proteins also generate Fe-S cofactors in the cytosol, using cytosolic isoforms of NFS1, ISCU etc. The hypothesis that nascent Fe-S clusters are assembled only in the

mitochondria and are then exported to the cytosol has been strongly challenged in multiple published papers, including some that they reference, and their discussion should be strengthened by acknowledging that an alternative Fe-S pathway for de novo Fe-S assembly is present in the cytosol. This pathway may even help them to interpret some of their data.

3. On page 5, critical for their maturation should be replaced with "critical for acquisition of the Fe-S cofactor. Most readers do not know what the term "maturation" means, and it is important that papers can be read by chemists.

4. Page 9, they left out some words- should be "bolstering its essentiality as EQUAL to that of E14.

5. On page 12, they tested iron binding, but they did not use informative units. They should represent as moles of iron per MOLE of protein. Then they can acknowledge that they showed much less iron binding than would be expected, considering that 4 iron atoms per protein is the expected ratio when the Fe-S has been fully reconstructed. Again, flawed protein "maturation" should be rephrased.

6. Page 17- used the word nearly, but seems that they meant "clearly".

7. In the competition assays of figure 6, it would be better to show competition of WT CDKAL1 against CDKAL1 with a mutagenized pentapeptide.

Overall, this paper is excellent and highly informative. It will provide a basis for understanding the important process of Fe-S acquisition in the many proteins that maintain genome integrity and likely lead to many other discoveries.

Referee #2:

The manuscript explores the poorly resolved question of how the CTC specifically recognizes 40 or more cytosolic and nuclear FeS proteins to insert their FeS cofactors. Overall, the authors uncover a consensus pentapeptide motif in a number of FeS proteins (that were known from earlier proteomic studies) as a specific CTC recognition signature for cluster delivery from CTC. This pentapeptide motif may complement the already known C-terminal tripeptide motif (containing a C-terminal W/F) present in a limited number of FeS proteins (see point 6 below).

The authors used the cytosolic FeS protein CDKAL1 (because of its simple domain structure and stability) to identify, by combined mutagenesis analyses and Ala scans, two small N-terminal regions that abolish core CTC (CIAO2B-CIAO1) binding. Only one region was further studied representing a pentapeptide that is conserved in both CDKAL1 proteins from different species and, more importantly, also found in some other FeS proteins (but see below). To verify the significance of this motif, the authors present a co-crystal structure of the Drosophila core CTC with a bound human pentapeptide DIEDI showing that the peptide sits at a cleft defined by the CIAO1-CIAO2B interface, comprising CIAO1 propeller tips and CIAO2B helices. Comparison with reference structures of the full CTC (Kassube) shows that MMS19 would bind opposite of the "client" binding site defined by the peptide, explaining its dispensable character for the interaction. Notably, the importance of R123-CIAO1 for pentapeptide interaction has been noted in earlier yeast work, and could be mentioned in support of the new findings (Srinivasan et al, 2007; doi 10.1016/j.str.2007.08.009). The importance of the individual residues within the pentapeptide was tested by pulldowns and ITC analyses to define an overall consensus motif termed CIM. Satisfactorily, the motif was also found in a number of other (known) FeS proteins. However, the authors should mention that this motif is also present in many other non-FeS proteins (Prosite search) and the (enthusiastic) statements that the peptide alone might allow the identification of new FeS proteins should be more cautiously phrased to avoid confusion of the reader.

The importance of the CIM motif in a number of FeS proteins for CTC binding was verified by mutagenesis and pulldowns. As a nice confirmation of the functional importance of the motif for these proteins, enzymatic assays, mainly from the DNA damage field, were performed. In particular, the functional dependence of the FANCD1 FeS protein was studied by several approaches, together showing that the CIM motif is important for FeS proteins involved in genome stability. Iron binding to the purified proteins was measured by a colorimetric (ferrozine) assay. From the results, I feel that the sensitivity of this assay for the amount of protein analyzed is at the detection limit, and hence, the approach may be questionable. In addition to Fig. 4A, the authors are requested to show the UV-Vis spectra of their proteins (easier and more convincing to show Fe+S binding, and doable with the amounts of proteins used). The wild-type proteins should have a nice brown color (420 nm peak) and the mutants should be colorless. What is the FeS cluster loading percentage of the studied FeS proteins? Is acid-labile sulfur detectable to quantitatively determine how much Fe and S is associated with the proteins?

Further, the authors reproduce the functional importance of the C-terminal tripeptide motif (W/F) for CTC binding reported earlier by other labs. They show that proteins with either penta- or tripeptide motifs can compete for binding to CTC. This is a bit surprising because it indicates high reversibility of binding, even though the complexes "survive" affinity purification. To verify and directly show the reversibility of CTC client binding, the authors should measure the dissociation of the binding partner during an extended wash in the affinity procedure. Finally, the authors use two client binding pocket mutants of CIAO1 and CIAO2B to analyze the differentially bound proteins (relative to wild-type). Which protein was used for the pull-down? If I get it right, two wild-type controls (for both CIAO1 and CIAO2B wild-type) are needed? An excel sheet of the proteomic data is missing to evaluate the specificity of the pull-down assay. This needs to be provided to get an idea of how many false-positives are co-purified in this test system and whether the FeS proteins are the top candidates. This would strengthen the view that both penta- or tripeptide motif-containing proteins use a similar region for CIAO1-CIAO2B binding, as suggested also by modeling. Overall, this nice and comprehensive study advances our knowledge of client protein recognition by the core CTC. Even though more interaction motifs may exist, as suggested in Fig. 6, the work presented here is impressive, carefully performed and may be of general interest, in particular for readers from metal biology, FeS biology, and the DNA damage field. A few amendments (see above and below) are requested to make the work even more compelling.

Specific comments

1. Abstract: the statement "AP-MS enabled us to explore the human CTC interactome" may be an exaggeration. It is "only" the CIM- (and C-term) specific CTC interactome what was identified.
2. Abstract: the statement "paved the way for expanding the repertoire of Fe-S proteins" likely needs to be tempered, as many proteins contain a CIM motif and it is not clear whether the sequence motif alone can help to extract new FeS proteins using the motif.
3. Introduction: The statement "Decades ago, these cofactors were discovered to be associated with an enzyme involved in DNA repair (Cunningham et al, 1989)." sounds as if this was the first discovered FeS protein. Of course, this is not true and FeS history goes back to the 60's (e.g., ferredoxins, respiratory chain). The authors may consider to rewrite this part appropriately.
4. Introduction: A statement that is made over and over again in the literature is that "Fe-S clusters are ironically prone to oxidative destruction in modern oxygen-rich environment." However, this is true only for a certain fraction of FeS proteins such as the cited hydratases (Imlay 2006) or regulatory proteins. Most other FeS clusters (in ferredoxins, respiratory chain and many other proteins) are firmly integrated into the holoproteins, and only few respond to oxidative stress, also in vivo. Please, correct this statement to reflect the complex nature of FeS protein stabilities.
5. Introduction: The citation for the statement "then exported to the cytosol for safe escort and presentation to the apo-proteins (Maio & Rouault, 2020)." is probably inappropriate, as T. Rouault denies the role of mitochondria for cytosolic FeS assembly. Another (or at least an additional) citation (from Lill and/or Dancis) might be fair to Rouault (even though not all share her view).
6. It would be nice to include a comment, whether the tripeptide W motif and the newly described DIEDI motif may occur also in a single protein. Further, it should be mentioned how many false positive examples for CIM are present in, e.g., the human genome.
7. Structurally speaking, have the authors analyzed whether the CIM is surface-exposed in the respective FeS proteins (if structure is available)?
8. Is it correct that many species contain a CDKAL1 protein without an N-terminal CIM? If correct, it would be informative to mention this.
9. In Fig. EV2E&F it would be better to mention the exact species name within the Figure, not only in the legend.
10. The explanation of the colors and circle size in Fig. 6E and EV6C needs to be better explained. What is exactly the difference? May be a more didactic version (or original data!!) is needed to present this data. Moreover, how does the IP assay account for false-positive pull-downs?
11. In Fig. 6G, please remove one of the two redundant cartoons of CTC-client complex (one per example is enough; I was confused thinking that the left and right cartoons in one box may be different).
12. Nomenclature: the term CIM may be confusing, since another "CIM" has been identified already by Marquez et al. 2023. Hence, I suggest to rename the motifs consistently as "TCR1" and "TCR2". This nomenclature would even be open for new motifs.

Reviewer #1

In this paper, the authors discover that a distinctive pentapeptide motif in proteins that acquire FeS cofactors in the cytosolic/nuclear compartment binds to a pocket in CIAO1 and its interface with CIA2. They use CDKAL1 for their experiments, as a representative, and they use sequence homology, alanine scanning, and affinity co-immunoprecipitations to prove their point. Then they examine other known FeS proteins in the cytosolic nuclear compartment and discover that the pentapeptide motif is present in many proteins of RNA and DNA metabolism. They model the pocket in the complex of CIAO1 and CIA2, and discover that proteins that acquire their FeS using a terminal W/F motif bind in the same pocket. Their work generates important insights into the molecular interactions that promote acquisition of critical FeS cofactors in many important proteins involved in genome maintenance, and their work also allows prospective identification of FeS candidate proteins that were not previously recognized. It is an excellent body of work that will energize FeS studies.

– We sincerely appreciate the reviewer's concise summary of our key findings and his/her high regard for our work.

There are several aspects of the paper that need correction or clarification:

1. In the abstract, they state that maturation of most eukaryotic Fe-S proteins requires the cytosolic Fe-S... On this point, it would be better to state something like- Most eukaryotic Fe-S proteins acquire their critical Fe-S cofactor by interacting with the cytosolic Fe-S cluster assembly targeting complex... Use of the word maturation is not universally accepted and description can be more precise, particularly in the abstract, which most people read before they commit to reading the paper.

– Although the term “maturation” has been frequently used in the literature to describe the process of Fe-S cluster delivery and insertion into apo-client Fe-S proteins that leads to the full functional capacity of these proteins, we completely agree with the reviewer that this term is not universally accepted and could be more precisely defined. To improve the clarity of our manuscript, we have revised the opening sentence of the abstract in accordance with the reviewer's suggestion (page 2). Furthermore, we have systematically avoided the use of “maturation” throughout the manuscript, replacing it with “cluster acquisition/delivery” or other context-appropriate alternatives.

2. They do not correctly represent results from papers written by the Rouault group. They need to read these papers and correctly assert that the Rouault group has found that the initial Fe-S assembly proteins also generate Fe-S cofactors in the cytosol, using cytosolic isoforms of NFS1, ISCU etc. The hypothesis that nascent Fe-S clusters are assembled only in the mitochondria and are then exported to the cytosol has been strongly challenged in multiple published papers, including some that they reference, and their discussion should be strengthened by acknowledging that an alternative Fe-S pathway for de novo Fe-S assembly is present in the cytosol. This pathway may even help them to interpret some of their data.

– We thank the reviewer for catching this mistake. We have revised the *Introduction* to include the *de novo* Fe-S assembly pathway in the cytosol and to acknowledge the parallel biogenesis of Fe-S clusters in both subcellular compartments (page 4). Consistent with this revision, we have also updated the references accordingly (page 4 and in the *References*; see also Reviewer #2 point 5).

3. On page 5, critical for their maturation should be replace with "critical for acquisition of the Fe-S cofactor. Most readers do not know what the term "maturation" means, and it is important that papers can be read by chemists.

– See also point 1. As noted above, to improve the clarity of our manuscript, we have replaced the term “maturation” throughout the manuscript with “cluster acquisition/delivery” or other context-appropriate alternatives (page 5 and other relevant sections).

4. Page 9, they left out some words- should be "bolstering its essentiality as EQUAL to that of E14.

– We've now added "equal to" to the sentence and rephrased it in line with the reviewer's suggestion (page 10).

5. On page 12, they tested iron binding, but they did not use informative units. They should represent as moles of iron per MOLE of protein. Then they can acknowledge that they showed much less iron binding that would be expected, considering that 4 iron atoms per protein is the expected ratio when the Fe-S has been fully reconstructed. Again, flawed protein "maturation" should be rephrased.

– We thank the reviewer for raising this critical point. After revising the representation of iron incorporation to the more informative unit of moles of iron per mole of protein (see revised Figure 4A), we noticed that, compared with our control FBXL5-SKP1-IRP2 complex containing a fully loaded [2Fe-2S] cluster (2 iron atoms per protein), the average iron-binding ratio of WT proteins was indeed lower than the expected value. We believe this insufficiency may arise from the limited cluster loading capacity of the mammalian cells utilized in our experiments, as well as partial oxidative damage during protein isolation. These considerations have now been included in the manuscript (page 13-14). In addition, we have supplemented the iron-binding assays with UV/Vis spectra (Figure EV3A), which further confirm the role of the pentapeptide motif in iron incorporation (see also Reviewer #2 general comments). Finally, the term "maturation" has been rephrased throughout the manuscript for greater precision (page 14).

6. Page 17- used the word nearly, but seems that they meant "clearly".

– We appreciate the reviewer catching this inappropriate phrasing and have now corrected it (page 18).

7. In the competition assays of figure 6, it would be better to show competition of WT CDKAL1 against CDKAL1 with a mutagenized pentapeptide.

– We thank the reviewer for this helpful suggestion, which would further support that the pentapeptide in CDKAL1 directly underlies its ability to compete with RSAD2 for CTC binding. We performed competitive bead-based affinity binding assays using both CDKAL1 WT and a CDKAL1 variant with an altered pentapeptide, as suggested (see revised text on page 17-18 and the results in Figure EV5D). Compared with the WT protein, the CDKAL1 variant with the altered pentapeptide lost its competitive capacity completely.

Overall, this paper is excellent and highly informative. It will provide a basis for understanding the important process of Fe-S acquisition in the many proteins that maintain genome integrity and likely lead to many other discoveries.

– Once again, we thank the reviewer for acknowledging the significance of our study and for the encouraging feedback.

Reviewer #2

The manuscript explores the poorly resolved question of how the CTC specifically recognizes 40 or more cytosolic and nuclear FeS proteins to insert their FeS cofactors. Overall, the authors uncover a consensus pentapeptide motif in a number of FeS proteins (that were known from earlier proteomic studies) as a specific CTC recognition signature for cluster delivery from CTC. This pentapeptide motif may complement the already known C-terminal tripeptide motif (containing a C-terminal W/F) present in a limited number of FeS proteins (see point 6 below).

The authors used the cytosolic FeS protein CDKAL1 (because of its simple domain structure and stability) to identify, by combined mutagenesis analyses and Ala scans, two small N-terminal regions that abolish core CTC (CIAO2B-CIAO1) binding. Only one region was further studied representing a pentapeptide that is conserved in both CDKAL1 proteins from different species and, more importantly, also found in some other FeS proteins (but see below). To verify the significance of this motif, the authors present a co-crystal structure of the Drosophila core CTC with a bound human

pentapeptide DIEDI showing that the peptide sits at a cleft defined by the CIAO1-CIAO2B interface, comprising CIAO1 propeller tips and CIAO2B helices. Comparison with reference structures of the full CTC (Kassube) shows that MMS19 would bind opposite of the "client" binding site defined by the peptide, explaining its dispensable character for the interaction. Notably, the importance of R123-CIAO1 for pentapeptide interaction has been noted in earlier yeast work, and could be mentioned in support of the new findings (Srinivasan et al, 2007; doi 10.1016/j.str.2007.08.009).

- We thank the reviewer for raising this point. In the revised manuscript, we have followed the advice and included this yeast study as an earlier result to support our findings about the role of human CIAO1 R125 in client recruitment (page 17 and in the *References*).

The importance of the individual residues within the pentapeptide was tested by pulldowns and ITC analyses to define an overall consensus motif termed CIM. Satisfactorily, the motif was also found in a number of other (known) FeS proteins. However, the authors should mention that this motif is also present in many other non-FeS proteins (Prosite search) and the (enthusiastic) statements that the peptide alone might allow the identification of new FeS proteins should be more cautiously phrased to avoid confusion of the reader.

- When conducting the Prosite search for the pentapeptide sequence, we indeed observed that many pentapeptide-containing proteins are non-Fe-S proteins, a point we have now clearly stated in the *Discussion* (page 22). We also fully agree with the reviewer that the presence of the pentapeptide or any sequence signature alone cannot warrant the identification of new Fe-S proteins. To prevent potential confusion for readers, we have revised the *Discussion* regarding the Prosite search as suggested (page 22) and have cautiously downplayed the emphasis on common feature decoding in the *Introduction* as well (page 4). We thank the reviewer for highlighting this critical point, which has helped us improve the clarity of our study.

The importance of the CIM motif in a number of FeS proteins for CTC binding was verified by mutagenesis and pulldowns. As a nice confirmation of the functional importance of the motif for these proteins, enzymatic assays, mainly from the DNA damage field, were performed. In particular, the functional dependence of the FANCI FeS protein was studied by several approaches, together showing that the CIM motif is important for FeS proteins involved in genome stability. Iron binding to the purified proteins was measured by a colorimetric (ferrozine) assay. From the results, I feel that the sensitivity of this assay for the amount of protein analyzed is at the detection limit, and hence, the approach may be questionable. In addition to Fig. 4A, the authors are requested to show the UV-Vis spectra of their proteins (easier and more convincing to show Fe+S binding, and doable with the amounts of proteins used). The wild-type proteins should have a nice brown color (420 nm peak) and the mutants should be colorless. What is the FeS cluster loading percentage of the studied FeS proteins? Is acid-labile sulfur detectable to quantitatively determine how much Fe and S is associated with the proteins?

- Owing to the difficulties in expressing and preparing sufficient amounts of Fe-S proteins from mammalian cells, the protein concentrations used for our iron-binding assays were at the lower end. To ensure the accuracy and reliability of these assays, we tested the linearity of the standard curve at low concentrations. Specifically, in addition to the typical 2, 4, 6, 8, and 10 nmole/well standards, we prepared a dilution series at 0.2, 0.4, 0.6, 0.8, and 1.0 nmole/well. Absorbance signals were measured, background-subtracted, and plotted as the standard curve (see below). Linear regression analysis yielded an R^2 value close to 1.0, confirming a strong linear relationship across all the standards and establishing a reliable quantification limit down to 0.2 nmole/well (absorbance \approx 0.017). Importantly, all tested sample readings fell within this quantifiable range (see Table below). To further validate the approach, we measured purified FBXL5-SKP1-IRP2 complex containing a fully loaded [2Fe-2S] cluster at two concentrations, both of which gave absorbance values comparable to our test samples (see Table below). According to the linear equation, the calculated iron-loading ratios for these control samples were very close to the expected 2 iron atoms per protein (see revised Figure 4A), supporting the reliability of our quantification.

By including the purified FBXL5-SKP1-IRP2 complex as a control and revising the representation of iron-binding assays to the unit of moles of iron per mole of protein (see revised Figure 4A), we noticed that the average iron-binding ratio of WT proteins tested was lower than the expected value. As mentioned in our response to Reviewer #1 (point 5) and now also clarified in the revised manuscript (page 13-14), this insufficiency may arise from the limited cluster loading capacity of the mammalian cells utilized in our experiments, as well as partial oxidative damage during protein isolation. To strengthen our observations, we have supplemented the iron-binding assays with UV/Vis spectra as requested (Figure EV3A). The spectra were displayed from 300 to 750nm for improved clarity. Due to the low yields and insufficient iron loading of the tested Fe-S proteins, the WT samples showed a modest bump, rather than a pronounced peak of characteristic absorbance at 420 nm, consistent with their faint coloration. By contrast, the mutants exhibited featureless spectra. These results have been incorporated into the revised manuscript (page 14; Figure EV3A).

Absorbance at 593 nm (A_{593})		
Proteins	WT	Mutant
FANCI	0.130	0.030
DNA2	0.146	0.029
MUTYH	0.185	0.032
TYW1	0.131	0.031
CDKAL1	0.162	0.031
	Sample 1	Sample 2
FBXL5-SKP1-IRP2	0.183	0.098

As to the acid-labile sulfur assay, we thank the reviewer for bringing up this classic and standard biochemical method for directly quantifying the inorganic sulfide content of Fe-S clusters. Unfortunately, due to technical issues and current laboratory limitations, we were unable to obtain reliable results at this stage. We are continuing to optimize the assay and plan to employ it in future work, particularly for the characterization of unrecognized Fe-S protein candidates.

Further, the authors reproduce the functional importance of the C-terminal tripeptide motif (W/F) for CTC binding reported earlier by other labs. They show that proteins with either penta- or tripeptide motifs can compete for binding to CTC. This is a bit surprising because it indicates high reversibility of binding, even though the complexes "survive" affinity purification. To verify and directly show the reversibility of CTC client binding, the authors should measure the dissociation of the binding partner during an extended wash in the affinity procedure.

- The reviewer has brought up an interesting point. While the results of our initial competition assay suggested a degree of binding reversibility, the primary purpose of this experiment was to demonstrate that both the pentapeptide and C-tail motifs engage the same pocket in the CTC when recruited. To clarify this and address the reviewer's concerns, we conducted additional bead-based competition assays, monitoring the dissociation of pre-bound clients in the flowthrough (we thank the reviewer for this helpful suggestion; see revised text on page 17 and results in revised Figure 6C). In these assays, freshly purified proteins were used with rigorous extended washes. As shown, both the pentapeptide-dependent client CDKAL1 and the C-tail-dependent client RSAD2 displayed dose-dependent competition for CTC binding without nonspecific "sticky" behavior. Consistent with this, CDKAL1 and RSAD2 exhibited comparable binding affinities to the core CTC complex as quantified by ITC (Figures EV1D and EV5C). Nonetheless, whether such binding reversibility and competition between CTC clients occur in cells remains to be determined.

In addition, to further prove that the pentapeptide in CDKAL1 directly accounts for its ability to compete with RSAD2 for CTC binding, we performed bead-based competition assays using both WT CDKAL1 and a pentapeptide-deficient variant (Figure EV5D; see also Reviewer #1, point 7). The results showed that, unlike the WT protein, the variant completely lost its competitive capacity.

Finally, the authors use two client binding pocket mutants of CIAO1 and CIAO2B to analyze the differentially bound proteins (relative to wild-type). Which protein was used for the pull-down? If I get it right, two wild-type controls (for both CIAO1 and CIAO2B wild-type) are needed? An xlsx sheet of the proteomic data is missing to evaluate the specificity of the pull-down assay. This needs to be provided to get an idea of how many false-positives are co-purified in this test system and whether the FeS proteins are the top candidates. This would strengthen the view that both penta- or tripeptide motif-containing proteins use a similar region for CIAO1-CIAO2B binding, as suggested also by modeling.

- We indeed appreciate the reviewer raising questions regarding the pocket-directed differential AP-MS. Due to space limitations in the manuscript, we did not elaborate on this experiment, which we believe is neat, brightly designed and, to a large degree, leads to the climax of our work. We take this opportunity here to provide a detailed explanation of the experiment and address the reviewer's questions.

As mentioned in our revised manuscript (page 22), after conducting a Prosite search for the pentapeptide and the C-tail sequence signatures, we obtained over 2,400 hits and realized that sequence signatures alone are insufficient for identifying hidden Fe-S proteins. To narrow down the candidates more precisely, we proceed to reveal a conserved pocket in the CTC dedicated to recruiting general clients (both pentapeptide-dependent and C-tail-dependent). Based on conserved and functionally critical residues within this pocket, we designed and performed pocket-directed differential AP-MS. This analysis involved curating and comparing results from three AP-MS experiments using purified CTC complexes as baits in the forms of WT (CIAO1-CIAO2B-MMS19 complex) and two pocket-deficient variants containing CIAO1-R125A (CIAO1^{R125A}-CIAO2B-MMS19 complex) and CIAO2B-R141A (CIAO1-CIAO2B^{R141A}-MMS19 complex) respectively. These single-site mutations have been shown to abolish sequence signature-dependent client recruitment by the revealed pocket (Figure 6B). During purification of the bait complexes, a STREP tag was left on CIAO1 for subsequent AP-MS. As the purified WT and variant CTC complexes were obtained in stoichiometry and displayed proper behavior, the differences observed among the three AP-MS datasets can, in principle, be attributed to pocket deficiency caused by the mutations. The *Methods* has been revised (page 36) to clearly specify the baits used in our AP-MS experiments.

In our revised manuscript, the results of the pocket-directed differential AP-MS are presented as: (1) a silver-stained gel showing the pull-down products of CTC-WT and two pocket-deficient CTC variants (Figure 6D), and (2) an Excel file containing the raw proteomic data from the three pull-down assays (Dataset EV1). We found that the majority of known CTC Fe-S protein clients, containing either the pentapeptide or the C-tail motif, along with their associated proteins, were readily detected in the CTC-WT spectra (though not always as top hits, due to differences in cellular abundance and intrinsic affinity for the CTC, e.g., TYW1 and POLE1), but were absent in both of the pocket-deficient spectra (Figure 6E; Dataset EV1 as highlighted). These observations validate the rationale of our pocket-directed differential AP-MS design and support its utility in uncovering hidden Fe-S proteins recruited via this pocket. We next handpicked hits localized to the cytoplasm or nucleus whose spectral counting pattern resembled those of known CTC clients (present in WT but diminished under both pocket-deficient conditions) as the candidates that are likely to contain unrecognized Fe-S cluster delivered by the CTC or their associating proteins (Figure EV5E; Dataset EV1 as highlighted). This pattern-based rather than spectral count-based selection turned out to be an effective experimental filter, allowing us to narrow the candidate list to approximately 50 proteins.

To specifically address the reviewer's questions, we used purified CTC complexes in the forms of WT and two pocket-deficient variants as baits for pull-down assays in order to investigate the CTC interactome in the context of the complete CTC complex rather than a single component. The MS results from the two pocket-deficient variants were compared with that of the WT complex, so only one control was required. The original proteomic data are provided in Dataset EV1, with protein hits of interest highlighted. As is common in typical AP-MS experiments, many non-Fe-S proteins co-purified as false positives and known Fe-S proteins were not always among the top hits due to differences in cellular abundance and intrinsic affinity for the CTC. However, our pocket-directed differential AP-MS strategy, combined with pattern-based rather than count-based selection largely circumvented the false-positive issue and yielded a compact, robust, and integrated list of candidates for further validation. For example, MATR3 and MAP7D3 showed consistently high counts in all AP-MS experiments and could have been mistaken as a candidate based on count alone. However, using our pattern-based selection across the three AP-MS datasets, they were excluded because they did not exhibit the characteristic spectral counting pattern of genuine CTC clients.

Overall, this nice and comprehensive study advances our knowledge of client protein recognition by the core CTC. Even though more interaction motifs may exist, as suggested in Fig. 6, the work presented here is impressive, carefully performed and may be of general interest, in particular for readers from metal biology, FeS biology, and the DNA damage field. A few amendments (see above and below) are requested to make the work even more compelling.

– We sincerely thank the reviewer for the encouraging comments and thoughtful suggestions, which have helped us to improve the clarity and impact of our study.

Specific comments

1. Abstract: the statement "AP-MS enabled us to explore the human CTC interactome" may be an exaggeration. It is "only" the CIM- (and C-term) specific CTC interactome what was identified.

– We agree with the reviewer that the statement "Subsequent structure-guided AP-MS enabled us to explore the human CTC interactome" may be somewhat overstated. What we actually did was, on top of identifying the pentapeptide, to further reveal a conserved pocket in the CTC dedicated to recruiting general clients (both pentapeptide-dependent and C-tail-dependent). Based on the residues within this pocket, we performed pocket-directed differential AP-MS analysis (see above for the detailed explanation of this experiment) to investigate the pocket-dependent human CTC interactome. To more faithfully reflect this, we have revised the statement to: "Subsequent structure-guided AP-MS enabled us to investigate the pocket-dependent human CTC interactome".

2. Abstract: the statement "paved the way for expanding the repertoire of Fe-S proteins" likely needs to be tempered, as many proteins contain a CIM motif and it is not clear whether the sequence motif alone can help to extract new FeS proteins using the motif.

– According to the reviewer's suggestion, we have replaced "paved the way" with "opened an avenue", which more accurately reflects the possibility, rather than certainty, of expanding the repertoire of Fe-S proteins.

3. Introduction: The statement "Decades ago, these cofactors were discovered to be associated with an enzyme involved in DNA repair (Cunningham et al, 1989)." sounds as if this was the first discovered FeS protein. Of course, this is not true and FeS history goes back to the 60's (e.g., ferredoxins, respiratory chain). The authors may consider to rewrite this part appropriately.

– The reviewer is absolutely correct. To avoid confusion, we have included a statement clarifying the history Fe-S protein studies and have revised the related part in the *Introduction* as appropriate (page 3 and in the *References*).

4. *Introduction: A statement that is made over and over again in the literature is that "Fe-S clusters are ironically prone to oxidative destruction in modern oxygen-rich environment." However, this is true only for a certain fraction of FeS proteins such as the cited hydratases (Imlay 2006) or regulatory proteins. Most other FeS clusters (in ferredoxins, respiratory chain and many other proteins) are firmly integrated into the holoproteins, and only few respond to oxidative stress, also in vivo. Please, correct this statement to reflect the complex nature of FeS protein stabilities.*

– We thank the reviewer for raising this point, and we have found it to be very enlightening. We agree with the reviewer that a good portion of Fe-S clusters are firmly integrated into the holoproteins and well-protected from environment factors, including oxidative stress and redox changes. The reviewer's comments remind us of the case of FBXL5, which we have dealt with: the Fe-S cluster in FBXL5 displayed high sensitivity to both oxidative and reductive disturbances when FBXL5 was isolated by itself. However, the cluster became very stable and resistant to these disturbances when FBXL5 was complexed with IRP2. Binding of IRP2 altered the local chemical environment of the FBXL5 Fe-S cluster localized close to the interface with IRP2, providing some degree of protection to the cluster. Therefore, we believe that, in addition to the integration property of Fe-S clusters, the local chemical environment also plays a role in mediating their sensitivity to oxidative and reductive disturbances.

To correctly reflect the complex nature of Fe-S protein stabilities, we have revised the relevant part in the *Introduction* and adjusted the references as appropriate (page 3-4, and in the *References*).

5. *Introduction: The citation for the statement "then exported to the cytosol for safe escort and presentation to the apo-proteins (Maio & Rouault, 2020)." is probably inappropriate, as T. Rouault denies the role of mitochondria for cytosolic FeS assembly. Another (or at least an additional) citation (from Lill and/or Dancis) might be fair to Rouault (even though not all share her view).*

– We thank the reviewer for catching this mistake and pointing out more appropriate references. We have now revised the *Introduction* to acknowledge the parallel biogenesis of Fe-S clusters in both mitochondria and the cytosol (page 4). Consistent with this revision and as suggested by the reviewer, we have also updated the references accordingly (page 4 and in the *References*; see also Reviewer #1 point 2).

6. *It would be nice to include a comment, whether the tripeptide W motif and the newly described DIEDI motif may occur also in a single protein. Further, it should be mentioned how many false positive examples for CIM are present in, e.g., the human genome.*

– The reviewer has raised an interesting point regarding the sequence signature. Among all known Fe-S proteins, none contain both a pentapeptide and a C-tail W/F motif. However, in the handpicked hits from our pocket-directed differential AP-MS, TRMT1L does possess both motifs, with DLENL as the pentapeptide and a C-terminal W. Which of these sequence signature is actually utilized for recruitment by the CTC remains to be determined. In line with this, we have included a comment on TRMT1L in the final part of the *Results* (page 19) and revised Figure 6F and EV5E accordingly.

For the false-positive examples of pentapeptide-containing proteins in the human proteome, as mentioned above, we were aware of this issue and have now clearly stated in the revised manuscript (page 22) that the presence of the pentapeptide alone is insufficient to identify new Fe-S proteins, as many such proteins are in fact non-Fe-S proteins. This limitation motivated us to identify the client-recruiting pocket in the CTC and to perform pocket-directed differential AP-MS for handpicking the final candidates. Among the approximately 1,200 pentapeptide-containing hits from the Prosite search, we indeed found numerous apparent non-Fe-S proteins, such as CDC26, LSM3, and PDC10, all of which lack cysteines and have structure solved without Fe-S cluster binding. However,

given the unknown number of hidden Fe-S proteins that contain pentapeptide, it is currently very difficult to estimate how many false-positive non-Fe-S proteins are present.

7. Structurally speaking, have the authors analyzed whether the CIM is surface-exposed in the respective FeS proteins (if structure is available)?

– We appreciate the reviewer bringing up this critical point that we are also very interested in. For pentapeptide-containing Fe-S proteins with structures covering this region available (including MUTYH, POLE1, DNA2, EXO5, and DPH3), we found that 4 out of 5 (with the exception of DNA2) display a flexible and solvent-exposed pentapeptide. For those without solved structures covering this region (including CDKAL1, KIF4A, TYW1, FBXL5, FANCI, and RTEL1), we applied AlphaFold3 predictions and observed that the pentapeptide regions in CDKAL1, KIF4A, TYW1, and FBXL5 tend to be solvent-exposed as well. These analyses suggest that the majority of the pentapeptide motifs are highly accessible and therefore well-suited for CTC recognition.

However, there are a few exceptions, such as structure-solved DNA2 harbors the pentapeptide within a well-folded strand, and FANCI and RTEL1 display this region as semi-solvent-exposed but likely inaccessible for interactions. In our opinion, prior to stably folding upon Fe-S cluster incorporation, these proteins may initially adopt a locally disordered and flexible conformation surrounding the Fe-S cluster binding and the pentapeptide regions (as in the case of XPD, Fan L *et al.* 2008 *Cell* 133(5):789-800), thereby permitting CTC recognition. Alternatively, co-translational Fe-S cluster insertion may provide another possibility to enable cluster acquisition when the recognition site/region is well-structured and inaccessible. As we are particularly interested in the dynamics of Fe-S cluster insertion and subsequent client release and have already initiated experiments in this respect, we did not elaborate on this point in the current manuscript.

8. Is it correct that many species contain a CDKAL1 protein without an N-terminal CIM? If correct, it would be informative to mention this.

– The reviewer is absolutely correct on this point. The bacterial CDKAL1 orthologs lack the pentapeptide motif we identified, consistent with the absence of the CTC complex and the reliance on alternative mechanisms for Fe-S cluster biogenesis and insertion in these species. We have now included this clarification in the revised manuscript (page 11). In addition, we updated Figures EV2E and EV2F to include several lower species such as placozoa and archaeon, providing more extensive sequence alignments.

9. In Fig. EV2E&F it would be better to mention the exact species name within the Figure, not only in the legend.

– As recommended, the exact species names have been included in the revised figures (Figures EV2E and EV2F).

10. The explanation of the colors and circle size in Fig. 6E and EV6C needs to be better explained. What is exactly the difference? May be a more didactic version (or original data!!) is needed to present this data. Moreover, how does the IP assay account for false-positive pull-downs?

– As requested by the reviewer, the original proteomic data are provided in Dataset EV1, with protein hits of interest highlighted. Also see above for a detailed explanation of our differential AP-MS design and pattern-based selection strategy. Figures 6E and EV5E present dot plots directly visualizing the results from the three AP-MS datasets. In both figures, the fill color represents the average spectral count of each protein, with darker shading indicating higher counts (maximum capped at 50; values above 50 are shown in black). The dot size reflects the relative abundance of each protein across the three datasets. Consistent with this, known pocket-dependent Fe-S proteins in Figure 6E show the largest dots under WT conditions, but very small or no dots under both pocket-deficient variants. A detailed explanation of the colors and circle size has been included in the revised *Figure Legends* for both figures.

To address the reviewer's concern about false-positive pull-downs in IP assays, we note that our pocket-directed differential AP-MS design inherently controls for this issue by comparing datasets from WT and two pocket-deficient variants, and by applying a pattern-based rather than count-based selection strategy to distinguish true pocket-dependent CTC clients from non-specific co-purified proteins, as detailed above.

11. In Fig. 6G, please remove one of the two redundant cartoons of CTC-client complex (one per example is enough; I was confused thinking that the left and right cartoons in one box may be different).

- We thank the reviewer for raising this point, which could indeed be confusing to readers. In Figure 6G, the left and right cartoons within each box illustrate distinct modes of recognition between the CTC and its clients: in the left cartoon, the sequence signature-containing client directly interacts with the CTC, whereas in the right cartoon, recognition is mediated by a sequence signature-containing adaptor protein (shown in brown) that bridges the client and the CTC (see also the mini graphical illustration in the bottom right corner). To improve clarity, we have added a statement in the revised *Figure Legends* to explicitly describe the differences between the two cartoons (page 48).

12. Nomenclature: the term CIM may be confusing, since another "CIM" has been identified already by Marquez et al. 2023. Hence, I suggest to rename the motifs consistently as "TCR1" and "TCR2". This nomenclature would even be open for new motifs.

- We thank the reviewer for pointing out the potential confusion caused by the term "CIM". We agree that readers may find it difficult to distinguish between the sequence motifs identified for CTC recognition based solely on the names "CIM" or "TCR". To prevent this confusion and emphasize the distinct feature of the sequence code we identified, we have decided to rename the motif more directly as "the Pentapeptide", in contrast to the C-tail tripeptide "TCR". Accordingly, we have replaced all the terms of "CIM" in the manuscript with "the Pentapeptide".

Dr. Hui Wang
Shenzhen Bay Laboratory
Greater Bay Biomedical InnoCenter
Shenzhen, Guangdong 518132
China

20th Oct 2025

Re: EMBOJ-2025-121548R
Client Recruitment Mechanism of the Cytosolic Fe-S Cluster Assembly Targeting Complex

Dear Dr. Wang,

Thank you for submitting your revised manuscript to The EMBO Journal. It has now been re-reviewed by the original referee 2, who was satisfied with most of the revisions, but retains several specific concerns. As you will see in the comments below, these include important issues with unsatisfactory answers to key questions, such as acid-lability assays, reversibility of client binding, as well as various presentational issues. I feel that in light of the potentially far-reaching implications of this work, these concerns would still need to be resolved, and therefore decided to return the manuscript to you for an exceptional second round of revision, which should hopefully allow you to adequately address referee 2's remaining points.

When preparing a re-revised manuscript, I would appreciate if you could also already incorporate the following editorial issues, which should greatly facilitate processing of the manuscript in case we should eventually proceed with its publication:

- Please adjust the order of the manuscript sections, and also make sure to use the correct section headers: Title page with complete author information, Abstract, Keywords, Introduction, Results, Discussion, Methods, Data Availability, Acknowledgements, Disclosure and Competing Interests Statement, References, Main Figure Legends, Tables, Expanded Figure Legends.
- Please carefully go through the reference list and make sure that each reference is complete with citation year, volume, and page/eLocator numbers - this information is currently missing for several of them.
- As we are switching from a free-text author contribution statement towards a more formal statement based on Contributor Role Taxonomy (CRediT) terms, please remove the present Author Contribution section and instead specify each author's contribution(s) directly in the Author Information page of our submission system during upload of the final manuscript. See <https://casrai.org/credit/> for more information.
- For Dataset EV1, please make sure that its legend/description is included (only) in a separate "Legend" tab of the XLSX spreadsheet.
- Please move the Reagents and Tools table from the main article file, and upload it as a separate text file. Also, please make sure to adhere to the template table downloadable from our author guidelines: <https://www.embopress.org/page/journal/14693178/authorguide#structuredmethods>
- During routine pre-acceptance checks, our data editors have raised the following queries regarding figures, data, and legends; I would appreciate if you briefly answered to them in the cover letter of your final submission, and made the requested text modifications with changes/additions highlighted via the "Track changes" option, to facilitate our final checking"
 1. Please note that the exact p values have to be provided in the legends of figures 4B-D
 2. Please note that the measure of center for the error bars needs to be defined in the legend of figure 4A
- Furthermore, our routine image checks revealed a white pixel line between lanes A7 and A8 in Figure 1D, which is not visible in the respective figure source data and therefore likely originates from digital conversion(s) of the data - to exclude misunderstandings, I would appreciate if you could re-assemble this figure before re-uploading.
- Finally, please provide suggestions for a short 'blurb' text prefacing and summing up the conceptual aspect of the study in two sentences (max. 250 characters), followed by 3-5 one-sentence 'bullet points' with brief factual statements of key results of the paper; they will form the basis of an editor-written 'Synopsis' accompanying the online version of the article. Please also upload a synopsis image, which can be used as a "visual title" for the synopsis section of your paper. The image (maybe based on Figure 6G?) should be in PNG or JPG format, and please make sure that it remains in the modest dimensions of (exactly) 550 pixels wide and 300-600 pixels high.

I am therefore returning the manuscript to you for a final round of revision, with the link below for eventual resubmission. Should

you have any questions regarding the referee comments or this decision, please do not hesitate to contact me directly.

Yours sincerely,

Hartmut Vodermaier

- 1) Every manuscript requires a Data Availability section (even if only stating that no deposited datasets are included). Primary datasets or computer code produced in the current study have to be deposited in appropriate public repositories prior to resubmission, and reviewer access details provided in case that public access is not yet allowed. Further information: embopress.org/page/journal/14602075/authorguide#dataavailability
- 2) Each figure legend must specify
 - size of the scale bars that are mandatory for all micrograph panels
 - the statistical test used to generate error bars and P-values
 - the type error bars (e.g., S.E.M., S.D.)
 - the number (n) and nature (biological or technical replicate) of independent experiments underlying each data point
 - Figures may not include error bars for experiments with $n < 3$; scatter plots showing individual data points should be used instead.
- 3) Revised manuscript text (including main tables, and figure legends for main and EV figures) has to be submitted as editable text file (e.g., .docx format). We encourage highlighting of changes (e.g., via text color) for the referees' reference.
- 4) Each main and each Expanded View (EV) figure should be uploaded as individual production-quality files (preferably in .eps, .tif, .jpg formats). For suggestions on figure preparation/layout, please refer to our Figure Preparation Guidelines: <http://bit.ly/EMBOPressFigurePreparationGuideline>
- 5) Point-by-point response letters should include the original referee comments in full together with your detailed responses to them (and to specific editor requests if applicable), and also be uploaded as editable (e.g., .docx) text files.
- 6) Please complete our Author Checklist, and make sure that information entered into the checklist is also reflected in the manuscript; the checklist will be available to readers as part of the Review Process File. A download link is found at the top of our Guide to Authors: embopress.org/page/journal/14602075/authorguide
- 7) All authors listed as (co-)corresponding need to deposit, in their respective author profiles in our submission system, a unique ORCID identifier linked to their name. Please see our Guide to Authors for detailed instructions.
- 8) Please note that supplementary information at EMBO Press has been superseded by the 'Expanded View' for inclusion of additional figures, tables, movies or datasets; with up to five EV Figures being typeset and directly accessible in the HTML version of the article. For details and guidance, please refer to: embopress.org/page/journal/14602075/authorguide#expandedview
- 9) To facilitate reproducibility and cross-laboratory adoption of methodologies, please structure the Materials & Methods section as outlined in our guide to authors, including a completed Reagents and Tools Table that can be downloaded from our author guidelines as well (<https://www.embopress.org/page/journal/14602075/authorguide#structuredmethods>).
- 10) Digital image enhancement is acceptable practice, as long as it accurately represents the original data and conforms to community standards. If a figure has been subjected to significant electronic manipulation, this must be clearly noted in the figure legend and/or the 'Materials and Methods' section. The editors reserve the right to request original versions of figures and the original images that were used to assemble the figure. Finally, we generally encourage uploading of numerical as well as gel/blot image source data; for details see: embopress.org/page/journal/14602075/authorguide#sourcedata

In the interest of ensuring the conceptual advance provided by the work, we recommend submitting a revision within 3 months (18th Jan 2026). Please discuss the revision progress ahead of this time with the editor if you require more time to complete the revisions. Use the link below to submit your revision:

Link Not Available

Referee #2:

Reviewer #2

Revised comments are marked with RR2:

The manuscript explores the poorly resolved question of how the CTC specifically recognizes 40 or more cytosolic and nuclear FeS proteins to insert their FeS cofactors. Overall, the authors uncover a consensus pentapeptide motif in a number of FeS proteins (that were known from earlier proteomic studies) as a specific CTC recognition signature for cluster delivery from CTC. This pentapeptide motif may complement the already known C-terminal tripeptide motif (containing a C-terminal W/F) present in a limited number of FeS proteins (see point 6 below).

The authors used the cytosolic FeS protein CDKAL1 (because of its simple domain structure and stability) to identify, by combined mutagenesis analyses and Ala scans, two small N-terminal regions that abolish core CTC (CIAO2B-CIAO1) binding. Only one region was further studied representing a pentapeptide that is conserved in both CDKAL1 proteins from different species and, more importantly, also found in some other FeS proteins (but see below). To verify the significance of this motif, the authors present a co-crystal structure of the Drosophila core CTC with a bound human pentapeptide DIEDI showing that the peptide sits at a cleft defined by the CIAO1-CIAO2B interface, comprising CIAO1 propeller tips and CIAO2B helices. Comparison with reference structures of the full CTC (Kassube) shows that MMS19 would bind opposite of the "client" binding site defined by the peptide, explaining its dispensable character for the interaction. Notably, the importance of R123-CIAO1 for pentapeptide interaction has been noted in earlier yeast work, and could be mentioned in support of the new findings (Srinivasan et al, 2007; doi 10.1016/j.str.2007.08.009).

- We thank the reviewer for raising this point. In the revised manuscript, we have followed the advice and included this yeast study as an earlier result to support our findings about the role of human CIAO1 R125 in client recruitment (page 17 and in the References).

RR2: Resolved

The importance of the individual residues within the pentapeptide was tested by pulldowns and ITC analyses to define an overall consensus motif termed CIM. Satisfactorily, the motif was also found in a number of other (known) FeS proteins. However, the authors should mention that this motif is also present in many other non-FeS proteins (Prosite search) and the (enthusiastic) statements that the peptide alone might allow the identification of new FeS proteins should be more cautiously phrased to avoid confusion of the reader.

- When conducting the Prosite search for the pentapeptide sequence, we indeed observed that many pentapeptide-containing proteins are non-Fe-S proteins, a point we have now clearly stated in the Discussion (page 22). We also fully agree with the reviewer that the presence of the pentapeptide or any sequence signature alone cannot warrant the identification of new Fe-S proteins. To prevent potential confusion for readers, we have revised the Discussion regarding the Prosite search as suggested (page 22) and have cautiously downplayed the emphasis on common feature decoding in the Introduction as well (page 4). We thank the reviewer for highlighting this critical point, which has helped us improve the clarity of our study.

RR2: Resolved

The importance of the CIM motif in a number of FeS proteins for CTC binding was verified by mutagenesis and pulldowns. As a nice confirmation of the functional importance of the motif for these proteins, enzymatic assays, mainly from the DNA damage field, were performed. In particular, the functional dependence of the FANCI FeS protein was studied by several approaches, together showing that the CIM motif is important for FeS proteins involved in genome stability. Iron binding to the purified proteins was measured by a colorimetric (ferrozine) assay. From the results, I feel that the sensitivity of this assay for the amount of protein analyzed is at the detection limit, and hence, the approach may be questionable. In addition to Fig. 4A, the authors are requested to show the UV-Vis spectra of their proteins (easier and more convincing to show Fe+S binding, and doable with the amounts of proteins used). The wild-type proteins should have a nice brown color (420 nm peak) and the mutants should be colorless. What is the FeS cluster loading percentage of the studied FeS proteins? Is acid-labile sulfur detectable to quantitatively determine how much Fe and S is associated with the proteins?

- Owing to the difficulties in expressing and preparing sufficient amounts of Fe-S proteins from mammalian cells, the protein concentrations used for our iron-binding assays were at the lower end. To ensure the accuracy and reliability of these assays, we tested the linearity of the standard curve at low concentrations. Specifically, in addition to the typical 2, 4, 6, 8, and 10 nmole/well standards, we prepared a dilution series at 0.2, 0.4, 0.6, 0.8, and 1.0 nmole/well. Absorbance signals were measured, background-subtracted, and plotted as the standard curve (see below). Linear regression analysis yielded an R² value close to 1.0, confirming a strong linear relationship across all the standards and establishing a reliable quantification limit down to 0.2 nmole/well (absorbance \approx 0.017). Importantly, all tested sample readings fell within this quantifiable range (see Table below). To further validate the approach, we measured purified FBXL5-SKP1-IRP2 complex containing a fully loaded [2Fe-2S] cluster at two concentrations, both of which gave absorbance values comparable to our test samples (see Table below). According to the linear equation, the calculated iron-loading ratios for these control samples were very close to the expected 2 iron atoms per protein (see revised Figure 4A), supporting the reliability of our quantification.

RR2: Resolved

By including the purified FBXL5-SKP1-IRP2 complex as a control and revising the representation of iron-binding assays to the unit of moles of iron per mole of protein (see revised Figure 4A), we noticed that the average iron-binding ratio of WT proteins tested was lower than the expected value. As mentioned in our response to Reviewer #1 (point 5) and now also clarified in the revised manuscript (page 13-14), this insufficiency may arise from the limited cluster loading capacity of the mammalian cells utilized in our experiments, as well as partial oxidative damage during protein isolation. To strengthen our observations, we have supplemented the iron-binding assays with UV/Vis spectra as requested (Figure EV3A). The spectra were displayed from 300 to 750nm for improved clarity. Due to the low yields and insufficient iron loading of the tested Fe-S proteins, the WT samples showed a modest bump, rather than a pronounced peak of characteristic absorbance at 420 nm, consistent with their faint coloration. By contrast, the mutants exhibited featureless spectra. These results have been incorporated into the revised manuscript (page 14; Figure EV3A).

RR2: The UV/Vis spectra are a nice confirmation for the FeS cluster presence. However, the data must be shown to include the 280 nm peak. The current figure parts can be converted to an insert. The 420/280 nm ratio will give the reader a good feeling of cluster loading, and I feel (and the authors admit that) that cluster loading is low.

As to the acid-labile sulfur assay, we thank the reviewer for bringing up this classic and standard biochemical method for directly quantifying the inorganic sulfide content of Fe-S clusters. Unfortunately, due to technical issues and current laboratory limitations, we were unable to obtain reliable results at this stage. We are continuing to optimize the assay and plan to employ it in future work, particularly for the characterization of unrecognized Fe-S protein candidates.

RR2: Unsatisfactory indeed. This may be due to the limited amounts of FeS on the various proteins. Usually, when Fe is detected, acid-labile sulfur can also be measured. Can the authors not take just high amounts of one FeS protein and show S content (by methylene blue)?

Further, the authors reproduce the functional importance of the C-terminal tripeptide motif (W/F) for CTC binding reported earlier by other labs. They show that proteins with either penta- or tripeptide motifs can compete for binding to CTC. This is a bit surprising because it indicates high reversibility of binding, even though the complexes "survive" affinity purification. To verify and directly show the reversibility of CTC client binding, the authors should measure the dissociation of the binding partner during an extended wash in the affinity procedure.

- The reviewer has brought up an interesting point. While the results of our initial competition assay suggested a degree of binding reversibility, the primary purpose of this experiment was to demonstrate that both the pentapeptide and C-tail motifs engage the same pocket in the CTC when recruited. To clarify this and address the reviewer's concerns, we conducted additional bead-based competition assays, monitoring the dissociation of pre-bound clients in the flowthrough (we thank the reviewer for this helpful suggestion; see revised text on page 17 and results in revised Figure 6C). In these assays, freshly purified proteins were used with rigorous extended washes. As shown, both the pentapeptide-dependent client CDKAL1 and the C-tail-dependent client RSAD2 displayed dose-dependent competition for CTC binding without nonspecific "sticky" behavior. Consistent with this, CDKAL1 and RSAD2 exhibited comparable binding affinities to the core CTC complex as quantified by ITC (Figures EV1D and EV5C). Nonetheless, whether such binding reversibility and competition between CTC clients occur in cells remains to be determined.

In addition, to further prove that the pentapeptide in CDKAL1 directly accounts for its ability to compete with RSAD2 for CTC binding, we performed bead-based competition assays using both WT CDKAL1 and a pentapeptide-deficient variant (Figure EV5D; see also Reviewer #1, point 7). The results showed that, unlike the WT protein, the variant completely lost its competitive capacity.

RR2: If I understand Fig. 6C correctly, the authors apparently have not exactly done what I suggested. Namely, using a non-equilibrium method such as extensive washing of a given column-bound CTC-client complex. Specifically, I suggested analyzing only a single CTC-client complex and monitoring the dissociation of the client over time. This would show that the competition

between different clients is a passive rather than active (where the incoming protein kicks out the bound one actively) mechanism. I am just puzzled that the client-CTC complex is so stable yet the exchange of clients works so efficiently. If this point will be resolved in later work, I am fine, but would suggest appropriate discussion of this issue.

Finally, the authors use two client binding pocket mutants of CIAO1 and CIAO2B to analyze the differentially bound proteins (relative to wild-type). Which protein was used for the pull-down? If I get it right, two wild-type controls (for both CIAO1 and CIAO2B wild-type) are needed? An xlsx sheet of the proteomic data is missing to evaluate the specificity of the pull-down assay. This needs to be provided to get an idea of how many false-positives are co-purified in this test system and whether the FeS proteins are the top candidates. This would strengthen the view that both penta- or tripeptide motif-containing proteins use a similar region for CIAO1-CIAO2B binding, as suggested also by modeling.

- We indeed appreciate the reviewer raising questions regarding the pocket-directed differential AP-MS. Due to space limitations in the manuscript, we did not elaborate on this experiment, which we believe is neat, brightly designed and, to a large degree, leads to the climax of our work. We take this opportunity here to provide a detailed explanation of the experiment and address the reviewer's questions.

As mentioned in our revised manuscript (page 22), after conducting a Prosite search for the pentapeptide and the C-tail sequence signatures, we obtained over 2,400 hits and realized that sequence signatures alone are insufficient for identifying hidden Fe-S proteins. To narrow down the candidates more precisely, we proceed to reveal a conserved pocket in the CTC dedicated to recruiting general clients (both pentapeptide-dependent and C-tail-dependent). Based on conserved and functionally critical residues within this pocket, we designed and performed pocket-directed differential AP-MS. This analysis involved curating and comparing results from three AP-MS experiments using purified CTC complexes as baits in the forms of WT (CIAO1-CIAO2B-MMS19 complex) and two pocket-deficient variants containing CIAO1-R125A (CIAO1R125A-CIAO2B-MMS19 complex) and CIAO2B-R141A (CIAO1-CIAO2BR141A-MMS19 complex) respectively. These single-site mutations have been shown to abolish sequence signature-dependent client recruitment by the revealed pocket (Figure 6B). During purification of the bait complexes, a STREP tag was left on CIAO1 for subsequent AP-MS. As the purified WT and variant CTC complexes were obtained in stoichiometry and displayed proper behavior, the differences observed among the three AP-MS datasets can, in principle, be attributed to pocket deficiency caused by the mutations. The Methods has been revised (page 36) to clearly specify the baits used in our AP-MS experiments.

In our revised manuscript, the results of the pocket-directed differential AP-MS are presented as: (1) a silver-stained gel showing the pull-down products of CTC-WT and two pocket-deficient CTC variants (Figure 6D), and (2) an Excel file containing the raw proteomic data from the three pull-down assays (Dataset EV1). We found that the majority of known CTC Fe-S protein clients, containing either the pentapeptide or the C-tail motif, along with their associated proteins, were readily detected in the CTC-WT spectra (though not always as top hits, due to differences in cellular abundance and intrinsic affinity for the CTC, e.g., TYW1 and POLE1), but were absent in both of the pocket-deficient spectra (Figure 6E; Dataset EV1 as highlighted). These observations validate the rationale of our pocket-directed differential AP-MS design and support its utility in uncovering hidden Fe-S proteins recruited via this pocket. We next handpicked hits localized to the cytoplasm or nucleus whose spectral counting pattern resembled those of known CTC clients (present in WT but diminished under both pocket-deficient conditions) as the candidates that are likely to contain unrecognized Fe-S cluster delivered by the CTC or their associating proteins (Figure EV5E; Dataset EV1 as highlighted). This pattern-based rather than spectral count-based selection turned out to be an effective experimental filter, allowing us to narrow the candidate list to approximately 50 proteins.

To specifically address the reviewer's questions, we used purified CTC complexes in the forms of WT and two pocket-deficient variants as baits for pull-down assays in order to investigate the CTC interactome in the context of the complete CTC complex rather than a single component. The MS results from the two pocket-deficient variants were compared with that of the WT complex, so only one control was required. The original proteomic data are provided in Dataset EV1, with protein hits of interest highlighted. As is common in typical AP-MS experiments, many non-Fe-S proteins co-purified as false positives and known Fe-S proteins were not always among the top hits due to differences in cellular abundance and intrinsic affinity for the CTC. However, our pocket-directed differential AP-MS strategy, combined with pattern-based rather than count-based selection largely circumvented the false-positive issue and yielded a compact, robust, and integrated list of candidates for further validation. For example, MATR3 and MAP7D3 showed consistently high counts in all AP-MS experiments and could have been mistaken as a candidate based on count alone. However, using our pattern-based selection across the three AP-MS datasets, they were excluded because they did not exhibit the characteristic spectral counting pattern of genuine CTC clients.

RR2: Thanks for this important clarification and addition. This tells the reader that the analyses are complex and need future dedicated studies to positively identify new FeS proteins and the role of the cluster. Point is resolved.

Overall, this nice and comprehensive study advances our knowledge of client protein recognition by the core CTC. Even though more interaction motifs may exist, as suggested in Fig. 6, the work presented here is impressive, carefully performed and may be of general interest, in particular for readers from metal biology, FeS biology, and the DNA damage field. A few amendments (see above and below) are requested to make the work even more compelling.

- We sincerely thank the reviewer for the encouraging comments and thoughtful suggestions, which have helped us to improve

the clarity and impact of our study.

Specific comments

1. Abstract: the statement "AP-MS enabled us to explore the human CTC interactome" may be an exaggeration. It is "only" the CIM- (and C-term) specific CTC interactome what was identified.
- We agree with the reviewer that the statement "Subsequent structure-guided AP-MS enabled us to explore the human CTC interactome" may be somewhat overstated. What we actually did was, on top of identifying the pentapeptide, to further reveal a conserved pocket in the CTC dedicated to recruiting general clients (both pentapeptide-dependent and C-tail-dependent). Based on the residues within this pocket, we performed pocket-directed differential AP-MS analysis (see above for the detailed explanation of this experiment) to investigate the pocket-dependent human CTC interactome. To more faithfully reflect this, we have revised the statement to: "Subsequent structure-guided AP-MS enabled us to investigate the pocket-dependent human CTC interactome".

RR2: The first sentence of the revised abstract is now misleading. Please rephrase to "Most "cytosolic and nuclear" eukaryotic Fe-S proteins acquire their critical Fe-S cofactor by interacting with the cytosolic Fe-S cluster assembly targeting complex (CTC)." Mitochondria contain almost 40% of the cellular FeS proteins and of course do not use CTC.

2. Abstract: the statement "paved the way for expanding the repertoire of Fe-S proteins" likely needs to be tempered, as many proteins contain a CIM motif and it is not clear whether the sequence motif alone can help to extract new FeS proteins using the motif.

- According to the reviewer's suggestion, we have replaced "paved the way" with "opened an avenue", which more accurately reflects the possibility, rather than certainty, of expanding the repertoire of Fe-S proteins.

RR2: Resolved

3. Introduction: The statement "Decades ago, these cofactors were discovered to be associated with an enzyme involved in DNA repair (Cunningham et al, 1989)." sounds as if this was the first discovered FeS protein. Of course, this is not true and FeS history goes back to the 60's (e.g., ferredoxins, respiratory chain). The authors may consider to rewrite this part appropriately.

- The reviewer is absolutely correct. To avoid confusion, we have included a statement clarifying the history Fe-S protein studies and have revised the related part in the Introduction as appropriate (page 3 and in the References).

RR2: Resolved

4. Introduction: A statement that is made over and over again in the literature is that "Fe-S clusters are ironically prone to oxidative destruction in modern oxygen-rich environment." However, this is true only for a certain fraction of FeS proteins such as the cited hydratases (Imlay 2006) or regulatory proteins. Most other FeS clusters (in ferredoxins, respiratory chain and many other proteins) are firmly integrated into the holoproteins, and only few respond to oxidative stress, also in vivo. Please, correct this statement to reflect the complex nature of FeS protein stabilities.

- We thank the reviewer for raising this point, and we have found it to be very enlightening. We agree with the reviewer that a good portion of Fe-S clusters are firmly integrated into the holoproteins and well-protected from environment factors, including oxidative stress and redox changes. The reviewer's comments remind us of the case of FBXL5, which we have dealt with: the Fe-S cluster in FBXL5 displayed high sensitivity to both oxidative and reductive disturbances when FBXL5 was isolated by itself. However, the cluster became very stable and resistant to these disturbances when FBXL5 was complexed with IRP2. Binding of IRP2 altered the local chemical environment of the FBXL5 Fe-S cluster localized close to the interface with IRP2, providing some degree of protection to the cluster. Therefore, we believe that, in addition to the integration property of Fe-S clusters, the local chemical environment also plays a role in mediating their sensitivity to oxidative and reductive disturbances.

To correctly reflect the complex nature of Fe-S protein stabilities, we have revised the relevant part in the Introduction and adjusted the references as appropriate (page 3-4, and in the References).

RR2: The differential situation is now properly explained. FBXL5 is a regulatory protein and the cluster needs to come on and off the "regulate" the iron metabolism. Resolved

5. Introduction: The citation for the statement "then exported to the cytosol for safe escort and presentation to the apo-proteins (Maio & Rouault, 2020)." is probably inappropriate, as T. Rouault denies the role of mitochondria for cytosolic FeS assembly. Another (or at least an additional) citation (from Lill and/or Dancis) might be fair to Rouault (even though not all share her view).

- We thank the reviewer for catching this mistake and pointing out more appropriate references. We have now revised the Introduction to acknowledge the parallel biogenesis of Fe-S clusters in both mitochondria and the cytosol (page 4). Consistent with this revision and as suggested by the reviewer, we have also updated the references accordingly (page 4 and in the References; see also Reviewer #1 point 2).

RR2: The exclusive role of mitochondria for general cytosolic and nuclear FeS protein biogenesis is documented by numerous studies that analyze the function of e.g. NFS1, FDX2, ISCU2, and frataxin. Clear functional data that a "backup" de novo ISC system exists in the eukaryotic cytosol has not been positively shown, and indeed has been refuted by biochemistry, cell biology, evolution biology (mitosomes), and the recently described mitochondrial GSH transporter phenotype. All these experiments fit to a mitochondria-localized FeS function of these early ISC proteins in extra-mitochondrial biogenesis. Even the Rouault group has published data consistent with this model (DOI 10.1074/jbc.M112.418889). What I meant is that the citation Maio & Rouault (2020) for the correct statement that mitochondria mature virtually all cellular proteins was wrong. I think it would be highly confusing to mention "an alternative de novo assembly system in the cytosol" on page 4.

6. It would be nice to include a comment, whether the tripeptide W motif and the newly described DIEDI motif may occur also in a single protein. Further, it should be mentioned how many false positive examples for CIM are present in, e.g., the human genome.

- The reviewer has raised an interesting point regarding the sequence signature. Among all known Fe-S proteins, none contain both a pentapeptide and a C-tail W/F motif. However, in the handpicked hits from our pocket-directed differential AP-MS, TRMT1L does possess both motifs, with DLENL as the pentapeptide and a C-terminal W. Which of these sequence signature is actually utilized for recruitment by the CTC remains to be determined. In line with this, we have included a comment on TRMT1L in the final part of the Results (page 19) and revised Figure 6F and EV5E accordingly.

For the false-positive examples of pentapeptide-containing proteins in the human proteome, as mentioned above, we were aware of this issue and have now clearly stated in the revised manuscript (page 22) that the presence of the pentapeptide alone is insufficient to identify new Fe-S proteins, as many such proteins are in fact non-Fe-S proteins. This limitation motivated us to identify the client-recruiting pocket in the CTC and to perform pocket-directed differential AP-MS for handpicking the final candidates. Among the approximately 1,200 pentapeptide-containing hits from the Prosite search, we indeed found numerous apparent non-Fe-S proteins, such as CDC26, LSM3, and PDC10, all of which lack cysteines and have structure solved without Fe-S cluster binding. However, given the unknown number of hidden Fe-S proteins that contain pentapeptide, it is currently very difficult to estimate how many false-positive non-Fe-S proteins are present.

RR2: Thanks for the additional explanation. Resolved

7. Structurally speaking, have the authors analyzed whether the CIM is surface-exposed in the respective FeS proteins (if structure is available)?

- We appreciate the reviewer bringing up this critical point that we are also very interested in. For pentapeptide-containing Fe-S proteins with structures covering this region available (including MUTYH, POLE1, DNA2, EXO5, and DPH3), we found that 4 out of 5 (with the exception of DNA2) display a flexible and solvent-exposed pentapeptide. For those without solved structures covering this region (including CDKAL1, KIF4A, TYW1, FBXL5, FANCL, and RTEL1), we applied AlphaFold3 predictions and observed that the pentapeptide regions in CDKAL1, KIF4A, TYW1, and FBXL5 tend to be solvent-exposed as well. These analyses suggest that the majority of the pentapeptide motifs are highly accessible and therefore well-suited for CTC recognition.

However, there are a few exceptions, such as structure-solved DNA2 harbors the pentapeptide within a well-folded strand, and FANCL and RTEL1 display this region as semi-solvent-exposed but likely inaccessible for interactions. In our opinion, prior to stably folding upon Fe-S cluster incorporation, these proteins may initially adopt a locally disordered and flexible conformation surrounding the Fe-S cluster binding and the pentapeptide regions (as in the case of XPD, Fan L et al. 2008 Cell 133(5):789-800), thereby permitting CTC recognition. Alternatively, co-translational Fe-S cluster insertion may provide another possibility to enable cluster acquisition when the recognition site/region is well-structured and inaccessible. As we are particularly interested in the dynamics of Fe-S cluster insertion and subsequent client release and have already initiated experiments in this respect, we did not elaborate on this point in the current manuscript.

RR2: Interesting new direction. Thanks for the explanations. Resolved

8. Is it correct that many species contain a CDKAL1 protein without an N-terminal CIM? If correct, it would be informative to mention this.

- The reviewer is absolutely correct on this point. The bacterial CDKAL1 orthologs lack the pentapeptide motif we identified, consistent with the absence of the CTC complex and the reliance on alternative mechanisms for Fe-S cluster biogenesis and insertion in these species. We have now included this clarification in the revised manuscript (page 11). In addition, we updated Figures EV2E and EV2F to include several lower species such as placozoa and archaeon, providing more extensive sequence alignments.

RR2: Important addition. Thanks. Resolved

9. In Fig. EV2E&F it would be better to mention the exact species name within the Figure, not only in the legend.

- As recommended, the exact species names have been included in the revised figures (Figures EV2E and EV2F).

RR2: Resolved

10. The explanation of the colors and circle size in Fig. 6E and EV6C needs to be better explained. What is exactly the difference? May be a more didactic version (or original data!!) is needed to present this data. Moreover, how does the IP assay account for false-positive pull-downs?

- As requested by the reviewer, the original proteomic data are provided in Dataset EV1, with protein hits of interest highlighted. Also see above for a detailed explanation of our differential AP-MS design and pattern-based selection strategy. Figures 6E and EV5E present dot plots directly visualizing the results from the three AP-MS datasets. In both figures, the fill color represents the average spectral count of each protein, with darker shading indicating higher counts (maximum capped at 50; values above 50 are shown in black). The dot size reflects the relative abundance of each protein across the three datasets. Consistent with this, known pocket-dependent Fe-S proteins in Figure 6E show the largest dots under WT conditions, but very small or no dots under both pocket-deficient variants. A detailed explanation of the colors and circle size has been included in the revised Figure Legends for both figures.

To address the reviewer's concern about false-positive pull-downs in IP assays, we note that our pocket-directed differential AP-MS design inherently controls for this issue by comparing datasets from WT and two pocket-deficient variants, and by applying a pattern-based rather than count-based selection strategy to distinguish true pocket-dependent CTC clients from non-specific co-purified proteins, as detailed above.

RR2: Resolved

11. In Fig. 6G, please remove one of the two redundant cartoons of CTC-client complex (one per example is enough; I was confused thinking that the left and right cartoons in one box may be different).

- We thank the reviewer for raising this point, which could indeed be confusing to readers. In Figure 6G, the left and right cartoons within each box illustrate distinct modes of recognition between the CTC and its clients: in the left cartoon, the sequence signature-containing client directly interacts with the CTC, whereas in the right cartoon, recognition is mediated by a sequence signature-containing adaptor protein (shown in brown) that bridges the client and the CTC (see also the mini graphical illustration in the bottom right corner). To improve clarity, we have added a statement in the revised Figure Legends to explicitly describe the differences between the two cartoons (page 48).

RR2: Thanks for the clarification. It is now clear. Resolved

12. Nomenclature: the term CIM may be confusing, since another "CIM" has been identified already by Marquez et al. 2023. Hence, I suggest to rename the motifs consistently as "TCR1" and "TCR2". This nomenclature would even be open for new motifs.

- We thank the reviewer for pointing out the potential confusion caused by the term "CIM". We agree that readers may find it difficult to distinguish between the sequence motifs identified for CTC recognition based solely on the names "CIM" or "TCR". To prevent this confusion and emphasize the distinct feature of the sequence code we identified, we have decided to rename the motif more directly as "the Pentapeptide", in contrast to the C-tail tripeptide "TCR". Accordingly, we have replaced all the terms of "CIM" in the manuscript with "the Pentapeptide".

RR2: Excellent idea. Thanks. Resolved

Reviewer #2

RR2: The UV/Vis spectra are a nice confirmation for the FeS cluster presence. However, the data must be shown to include the 280 nm peak. The current figure parts can be converted to an insert. The 420/280 nm ratio will give the reader a good feeling of cluster loading, and I feel (and the authors admit that) that cluster loading is low.

- As requested, the UV/Vis spectra in Figure EV3A now display the wavelength range from 250 to 750 nm, allowing the 280 nm peak used for cluster-loading estimation to be clearly visualized. For clarity, the zoom-in view of the characteristic 420 nm absorbance has been converted into an inset within each spectrum. The corresponding figure legend has also been updated to reflect these changes.

RR2: Unsatisfactory indeed. This may be due to the limited amounts of FeS on the various proteins. Usually, when Fe is detected, acid-labile sulfur can also be measured. Can the authors not take just high amounts of one FeS protein and show S content (by methylene blue)?

- Following the reviewer's suggestion, we quantified the inorganic sulfide (S^{2-}) released from Fe-S clusters using a representative sample of CDKAL1 with relatively high protein yield. The acid-labile sulfide content was measured via methylene blue formation, as previously described (Rabinowitz JC 1978 *Methods Enzymol* 53:275-277). Briefly, sulfide (H_2S) reacts with N,N-dimethyl-p-phenylenediamine (DMPD) and excess $FeCl_3$, generating methylene blue, which we detected by its absorbance at ~ 670 nm. We proportionally downscaled this reaction system and performed the measurement in a 96-well plate using a 200 μ l sample volume.

Considering the low cluster-loading efficiency observed in our iron incorporation assays, we generated a standard curve using freshly prepared Na_2S at concentrations ranging from 0 to 10.0 nmole/well, with higher density at the lower end (0, 0.5, 1.0, 1.5, 2.0, 4.0, 6.0, 8.0, 10.0). After measuring and background-subtracting the absorbance, linear regression analysis yielded an R^2 value close to 1.0, confirming a strong linear relationship across all the standards (see below). The mean absorbance values from three replicates were 0.118 for CDKAL1 WT and 0.035 for its pentapeptide-deficient mutant, both within the quantifiable range. Based on the standard curve, the calculated sulfide content was 1.634 ± 0.072 and 0.425 ± 0.037 moles/mole of protein for the WT and mutant, respectively. These values are comparable to the iron-loading data previously obtained from an independent protein prep for CDKAL1 WT and mutant. These new results have been included and discussed in the updated manuscript (see revised text on page 13-14 and in the *Methods*).

Due to the low and variable Fe-S cluster content among different isolated Fe-S proteins, we have focused on quantifying acid-labile sulfide in CDKAL1 WT and its mutant as a representative case. Using the classic methylene blue assay, we successfully established this method as a feasible and informative complement to iron-binding measurements. We appreciate the reviewer's suggestion to incorporate this standard biochemical approach and hope that our new results satisfactorily address his/her concerns.

RR2: If I understand Fig. 6C correctly, the authors apparently have not exactly done what I suggested. Namely, using a non-equilibrium method such as extensive washing of a given column-bound CTC-client complex. Specifically, I suggested analyzing only a single CTC-client complex and monitoring the dissociation of the client over time. This would show that the competition between different clients is a passive rather than active (where the incoming protein kicks out the bound one actively) mechanism. I am just puzzled that the client-CTC complex is so stable yet the exchange of clients works so efficiently. If this point will be resolved in later work, I am fine, but would suggest appropriate discussion of this issue.

- To directly address the reviewer's concern regarding passive client dissociation during extensive washing, we immobilized the pre-assembled CTC-RSAD2 complex on affinity columns and performed both passive (buffer-only) and active competition assays (buffer supplemented with 5 μ M CDKAL1). To allow sufficient time for potential dissociation or displacement, the entire process was extended to 50 column volumes (CV), which took approximately 2 hours at a slow, fixed flow rate (~0.08 ml/min). We then monitored RSAD2 release in the flowthrough under both conditions.

These new results have been included and discussed in the updated manuscript (Figure EV5D; see revised text on page 18 and in the *Methods*). As shown, RSAD2 displayed only marginal passive dissociation from the pre-assembled CTC-client complex during prolonged washing, whereas the presence of 5 μ M CDKAL1 led to a significantly higher level of RSAD2 displacement. Thus, the client exchange observed in Figure 6C is predominantly attributable to active competition rather than passive dissociation.

We truly appreciate the reviewer for raising this insightful question. It was a bit mind-blowing to observe client dissociation from the CTC-client complex, even at a barely detectable level, during the extensive washing process. This experiment proved highly informative, offering valuable insights into the biochemical properties and dynamic nature of protein-protein interactions within this system.

RR2: The first sentence of the revised abstract is now misleading. Please rephrase to "Most "cytosolic and nuclear" eukaryotic Fe-S proteins acquire their critical Fe-S cofactor by interacting with the cytosolic Fe-S cluster assembly targeting complex (CTC)." Mitochondria contain almost 40% of the cellular FeS proteins and of course do not use CTC.

- As the reviewer correctly noted, the CTC functions specifically for cytosolic and nuclear Fe-S proteins. We thank the reviewer for pointing this out and have rephrased the sentence accordingly (page 2).

RR2: The exclusive role of mitochondria for general cytosolic and nuclear FeS protein biogenesis is documented by numerous studies that analyze the function of e.g. NFS1, FDX2, ISCU2, and frataxin. Clear functional data that a "backup" *de novo* ISC system exists in the eukaryotic cytosol has not been positively shown, and indeed has been refuted by biochemistry, cell biology, evolution biology (mitosomes), and the recently described mitochondrial GSH transporter phenotype. All these experiments fit to a mitochondria-localized FeS function of these early ISC proteins in extra-mitochondrial biogenesis. Even the Rouault group has published data consistent with this model (DOI 10.1074/jbc.M112.418889). What I meant is that the citation Maio & Rouault (2020) for the correct statement that mitochondria mature virtually all cellular proteins was wrong. I think it would be highly confusing to mention "an alternative *de novo* assembly system in the cytosol" on page 4.

- We appreciate the reviewer for clarifying this point and agree that the presence of a *de novo* Fe-S assembly system in the cytosol remains a highly controversial topic in the field. To prevent any potential misunderstanding, we have removed the phrase "an alternative *de novo* assembly system in the cytosol" from the revised manuscript and updated the references accordingly (page 4 and in the *References*).

Dr. Hui Wang
Shenzhen Bay Laboratory
Greater Bay Biomedical InnoCenter
Shenzhen, Guangdong 518132
China

12th Dec 2025

Re: EMBOJ-2025-121548R1
Client Recruitment Mechanism of the Cytosolic Fe-S Cluster Assembly Targeting Complex

Dear Dr. Wang,

Thank you for submitting your final revised manuscript for our consideration. I am pleased to inform you that we have now accepted it for publication in The EMBO Journal.

You may qualify for financial assistance for your publication charges - either via a Springer Nature fully open access agreement or an EMBO initiative. Check your eligibility: <https://link.springer.com/journal/44318/how-to-publish-with-us>

Yours sincerely,

Hartmut Vodermaier

Please note that it is The EMBO Journal policy for the transcript of the editorial process (containing referee reports and your response letters) to be published as an online supplement to each paper. If you should prefer removal of any referee-only figures included in the point-by-point response(s), e.g. because they may still be used for future publication or because they have been reproduced from published work by others, please do let us know immediately via response email.

More information is available here: <https://link.springer.com/partners/embo-press/editorial-policies#Peer%20review>